# *Verbum Verbo Concepisti*. The Word's Incarnation in Some Images of the Annunciation in the Light of Medieval Liturgical Hymns

José María Salvador-González

Art History Department, Faculty of Geography and History, Complutense University of Madrid, Campus Moncloa, 20040 Madrid, Spain; jmsalvad@ucm.es

**Abstract:** This article aims to explain why, in some European representations of the Annunciation, a bundle of rays of light comes from the mouth of God the Father toward the head/ear of the Virgin Mary. In order to find a satisfactory answer to this problem, the author first studies a series of biblical, patristic, theological, and liturgical sources referring to the supernatural human conception of the Word of God in Mary's immaculate womb. He then analyzes eleven images of the Annunciation that present this peculiarity. Finally, through a comparative analysis between the doctrinal texts and these exceptional images, the author concludes that the latter illustrate as visual metaphors the textual metaphors contained in the writings of some Church Fathers, medieval theologians, and liturgical hymnographers; that is to say, the beam of rays of light emitted by the mouth of the Most High to the Virgin's head/ear metaphorizes the human conception/incarnation of the Word of God in the virginal womb of Mary.

**Keywords:** Christology; Mariology; divine motherhood; Christ's conception; medieval culture; symbol; Marian iconography

## 1. Introduction

Throughout an extensive research on the iconography of the Annunciation to Mary, the author soon found that, since the end of the thirteenth century, a practically essential element is a beam of rays of light that, coming from God the Father—sometimes invisible or iconically figured—descends toward the Virgin, often carrying in its wake the Holy Spirit shaped as a dove. The author has shown in another study (Salvador-González 2022, pp. 334–55) that this ray of light is the symbolic representation of God the Son who, coming from God the Father, goes to the Virgin to be humanly incarnated in her immaculate womb.

Furthermore, this identification of Christ as a ray of light in the *Annunciations* was predictable if we remember that the Messiah identified himself as "the light of the world". This is how John relates it in his Gospel: "When Jesus spoke again to the people, he said, 'I am the light of the world. Whoever follows me will never walk in darkness, but will have the light of life'". (John 8:12. Bible online. New International Version).[1]

Now, in a careful study of the images of the Annunciation, the author was intrigued to see in some of them the surprising detail that this bundle of rays of light goes out from the mouth of God the Father, as if he were exhaling a breath or pronouncing a word. It did not take long for us to remember that Christ is defined by John in his Gospel as the Word of God (*Verbum Dei*), or the Reason (λόγος) of God. This is what the evangelist proclaims when referring to God the Son in these terms:

> In the beginning was the Word, and the Word was with God, and the Word was God. He was with God in the beginning. Through him all things were made;

without him nothing was made that has been made. In him was life, and that life was the light of all mankind. The light shines in the darkness, and the darkness has not overcome it. (John 1:1–5. Bible online. New International Version).[2]

Furthermore, since the author is dealing with a ray of light emitted by the mouth of the eternal Father, as if it were a "breath" or a "word," in images of the Annunciation, it is worth remembering that, at the end of this Marian episode, the Virgin manifests her unrestricted obedience to the plan of the Most High with this exclamation: "'I am the Lord's servant,' Mary answered. 'May your word to me be fulfilled'". (Luke 1:38. Bible online. New International Version).[3] It is important to emphasize in this regard that the Church Fathers and the theologians interpreted this sentence of Mary as referring not to the word of Gabriel but to the word of God the Father, transmitted to Mary by the archangel.

Given this background, the problem posed by this beam of rays of light emitted by the mouth of the eternal Father in some images of the Annunciation prompted us to seek a satisfactory answer through a threefold methodological approach: first, searching through primary sources of Christian doctrine, in quest of patristic, theological and hymnographic texts referring to the Word of God, whose analysis would shed light on this iconographic problem; second, an exploration of images of the Annunciation, to record and analyze those cases in which one can see that the beam of light is emitted by the mouth of God the Father; third, as a conclusion, a comparative analysis between these doctrinal texts and these very special *Annunciations*, to see if, and to what extent, the latter illustrate as visual metaphors the textual metaphors contained in the writings of the Fathers, medieval theologians and liturgical hymnographers.

As it is well known, practically all the great masters of Christian doctrine in the East and West have written about Christ in his status as the Word of God incarnated as man. Thus, attempting to present in this article this vast amount of patristic exegesis on the topic in question is impossible. Therefore, in Section 2, the author will refrain from presenting only a few testimonies of Eastern and Western Church Fathers and theologians on the subject. However, in Section 3 he will focus his attention *in extenso* on the medieval liturgical hymns that exalt Christ as the Word of God and, additionally, as the Light of the world.

## 2. Reflections on the Word of God in Some Fathers and Medieval Theologians

In this regard, the author can cite the testimonies of three influential masters as representative of the Fathers of the Greek-Eastern Churches.

St. Proclus, Patriarch of Constantinople from 434 to 446 (the year of his death), states in a sermon on Easter that the phrase in the Gospel of John "In the beginning was the Word" expresses not that the Word was made, but that he was at the beginning. Furthermore, the person of the Son was eternal and related to God the Father; therefore, he says that "he was with God. And the Word was God".[4] According to this prelate, the evangelist emphasizes that between God the Father and the Word of God there subsists an identical divine nature, an equal coeternity, and an identical power to create the universe.[5] In another later paragraph of this homily, Proclus asserts that when John says in his Gospel, "The Word became flesh", he reveals that the Word was incarnated on earth without ever leaving heaven. Furthermore, the Virgin Mary conceived him without the mediation of a man and without losing her virginity and gave birth to him in a painless delivery.[6]

Almost three centuries later, the famous hymnographer Saint Andrew of Crete, Archbishop of Gortyna, in Crete (c. 650–c. 712/40), points out in a hymn to the birth of the Virgin Mary that she was made a golden censer because in her womb the Word of fire would dwell by the work of the Holy Spirit, and from her, as a virginal mother, the human form of the divine Word would appear.[7] In another poetic composition, Andrew of Crete states that the Word became flesh and the Virgin gave birth to God made man, whom the star announces,

the Magi adore, the shepherds admire him, and nature exults in him. Therefore–concludes the author–Mary is the Virgin Mother of God, because she gave birth to the Savior and, with the Father's approval, gestated in her womb the Word of God clothed in flesh.[8]

Some years later, the prolific theologian St. John Damascene (675–749) affirmed in a homily on Holy Saturday that the sentence from the Gospel of John "The Word became flesh and dwelt among us" was made effective in the Virgin Mary. Moreover, she preserved her virginity after childbirth in a way as inexplicable as the birth of the Word of God incarnated as man is inexplicable.[9] Because of this human incarnation—Damascene goes on to explain—the invisible and impassible Word of God becomes visible and passible through his human flesh, and preserving his divinity consubstantial with that of the Father, the Word assumes our human substance, equal to ours in everything, except sin. The Virgin Mary is the mother of the Word of God because she conceived him without human semen and preserved her virginity.[10]

In a similar way to what you have seen in the Greek-Eastern sphere, one can highlight in the Latin Church, as representative of the subject under analysis some fragments of writings by the following six masters:

In the second half of the fourth century, St. Ambrose, Archbishop of Milan (339/40-397), composed the famous hymn *Veni Redemptor gentium*, in two of whose stanzas he proclaims:

| | |
|---|---|
| The Word of God became flesh | Non ex virili semine |
| Not by manly semen, | Sed mystico spiramine |
| But by a mystical breath, | Verbum Dei factum est caro, |
| And the fruit of the womb blossomed. | Fructusque ventris floruit. |
| The womb of the Virgin swells, | Alvus tumescit virginis, |
| The seal of virginity remains, | claustrum pudoris permanet, |
| The banners of virtues shine, | vexilla virtutum micant: |
| God dwells in the temple. | versatur in templo Deus.[11] |

About three generations later, St. Maximus, Bishop of Turin († c. 465), in a homily on Christ's Nativity, begins by quoting the aforementioned passage from the Gospel of John, stating that the Word existed at the beginning with God and was God and then became flesh.[12] Maximus goes on to say that God the Son's divinity comes from God the Father, while his flesh comes from a woman. Thus, the Word became flesh, not in the sense that God emptied himself into man, but in the sense that man was glorified in God.[13]

About seven centuries later, St. Anselm of Aosta, Archbishop of Canterbury (1033–1109), proclaims in a poem in honor of the Virgin Mary:

| | |
|---|---|
| Hail, mother, whose birth, | Ave mater, cujus partus |
| A torrent of supreme pleasure, | Torrens summae voluptatis |
| Is the Word of the Father made man, | Verbum Patris homo factus, |
| Fountain of life, splendor of light. | fons vitae est splendor lucis.[14] |

Some five decades later, the Cistercian St. Bernard, Abbot of Clairvaux (1090–1153), interpreting in his fourth homily *Super missus est* the testimony of Mary "*fiat mihi secundum verbum tuum*", with which she unconditionally accepted the divine plan at the final moment of the Annunciation, expresses suggestively, after assuming the role of the Virgin, and playing with the homologous terms "Word" (of God) and "word":

> Let it be made in me on the Word [of God] according to your word. The Word
> that was at the beginning with God be made flesh of my flesh according to your

word. Let it be made in me, I pray, the Word not uttered, passing away [without a trace], but conceived to remain, that is, clothed in flesh, not in air. Let it be made in me not only audible to the ears, but also visible to the eyes, palpable to the hands, portable to the shoulders.[15]

The Abbot of Clairvaux then metaphorically enhances the differences between the living Word and the inert word, oral and written, through these poetic concepts:

And do not make in me a written and mute word, but incarnate and living [Word]; that is, not with mute figures, engraved on dead parchments, but in human form vividly imprinted on my chaste viscera: and that not with a figuration of a dead pen, but by the operation of the Holy Spirit.[16]

The Cistercian Saint Amadeus, Bishop of Lausanne (1110–1159), states in a homily on Christ's incarnation that the Word descended below himself when he became incarnate and dwelt among us, humiliating himself by taking the form of a slave. That humiliation—the Swiss prelate goes on—was His human descent to earth, in which, however, by becoming flesh, he did not cease to be God, and the appropriation of is humanity did not diminish the glory of his divine majesty.[17]

More than a century later, Saint Bonaventure, Minister General of the Order of Friars Minor (c. 1217/21-1274), in his second sermon on Christ's Nativity, asserts that the phrase from the Gospel of John "The Word became flesh" expresses the heavenly mystery and the admirable sacrament, the magnificent work, and the infinite benefit of the eternal God, humbly bending down, assuming the clay of our nature, uniting it to the divine person. Therefore, the names Word and flesh are interconnected.[18] A few paragraphs later in this sermon, Bonaventure plays poetically with the analogies between the incarnate Word and the mental word when he asserts:

For when the mental word is spoken outwardly, it is clothed as if with a voice, and so it continues, it sounds in public so that it remains signified in the hidden, because the voice is perceived with the sense, but its meaning is perceived with the intellect. But the Word of the Father was certainly at first naked, because it is not joined to any creature; but afterwards, clothed with flesh, he showed the flesh to the outside, hiding the Divinity within; Isaiah 45: *Truly you are the hidden God*. Note also that the word of the mind and the word of the voice are not two words, but one, at first certainly naked, then clothed. So also the Word-flesh, being God and man, are not two, but one, Christ.[19]

After this brief presentation of some representative examples of patristic and theological testimonies on the subject under study, it is time to focus on a set of liturgical chants that address it poetically.

## 3. Poetic Speculations on the Incarnate Word of God in Medieval Liturgical Hymns

In this Section 3, the author will concentrate on studying numerous stanzas of Latin medieval liturgical hymns that exalt Christ as the incarnate Word of God or also as the Light of the World. To better appreciate the similarities and/or possible conceptual variations in the treatment of the theme over time, he will present these hymns in strict chronological order, grouping them by century.

### 3.1. Sixth-Century Hymn

In the sixth century, Ennodius, Bishop of Pavia (c. 473/74-521), extols in his *Hymnus 63. (10.) Hymnus sanctae Mariae* the incarnation of the divine Word in Mary's womb by the insufflation of the Spirit of God through this suggestive expression:

| | |
|---|---|
| The belly swells with the Spirit, | Turgescit alvus spiritu, |
| What the tongue released is semen, | Quod lingua iecit, semen est, |
| The Word is restricted in flesh, | In carne Verbum stringitur, |
| He is all ours through his mother. | De matre cunctus noster est.[20] |

### 3.2. Ninth-Century Hymn

In the ninth century, Rabanus Maurus, Archbishop of Mainz (c. 776/80-856), expressed in his *Hymnus 145. (14.) Item de Nativitate Domini* the supernatural birth of the divine Redeemer in these terms:

| | |
|---|---|
| The Word of the Father made flesh | Verbum patris caro factum |
| Is born of the Virgin, | nascitur ex virgine, |
| Without losing his divinity | Non amissa deitate |
| He takes the form of a servant, | formam servi suscipit, |
| So that, as Omnipotent, | Ut peccatum de peccato |
| He might destroy sin from sin. | damnaret omnipotens.[21] |

### 3.3. Tenth-Century Hymns

Dated in the tenth century, the author has documented these four hymns referring to the topic under consideration:

*Hymnus 2. De Adventu Domini Nostri* urges the Virgin to receive into her womb the Word of God from the mouth of God the Father in these explicit terms:

| | |
|---|---|
| Blessed Virgin Mary, | Verbum salutis omnium |
| receive in your chaste womb | Patris ab ore prodiens, |
| the Word of universal salvation | Virgo beata, suscipe |
| that proceeds from the mouth of the Father. | Casto Maria viscere.[22] |

*Hymnus 11. De Annuntiatione Beatae Mariae Virginis* glorifies the Virgin for her exclusive privilege of engendering the Word of God as man by stating:

| | |
|---|---|
| Hail, hail, full of grace, | Ave, ave, gratia plena, |
| The Lord is with you, | Dominus tecum, |
| You will conceive and give birth | Concipies et paries |
| To the Word of the Most High Father; | Verbum patris altissimi; |
| And the Word became flesh, | Et verbum caro factum est, |
| He is with us | Nobiscum est, |
| And remains eternally. | Et manet in aeternum.[23] |

*Hymnus 6. Purificatio* exalts Mary as the virgin mother of the Son of God made man through these simple rhymes:

| | |
|---|---|
| You conceived with honor | Concepisti suave |
| The gentle Word, | Verbum cum honore, |
| Taking that "Hail" | Sumens illud Ave |
| From the mouth of Gabriel. | Gabrielis ore.[24] |

*Hymnus 87. De Nominibus Domini* praises Christ with several lyrical metaphors, including "Word" and "light," in these brief verses:

| Mouth, Word, splendor, sun, glory, light | Os, verbum, splendor, sol, gloria, lux et |
| And image, | imago, |
| Bread, flower, vine, mountain, door, rock | Panis, flos, vitis, mons, ianua, petra |
| and stone. | lapisque.[25] |

### 3.4. Eleventh-Century Hymns

Dated to the eleventh century, we have recorded the following three hymns:

*Hymnus 9. In Epiphania Domini* announces the human incarnation of the Word/Light in the womb of Mary, stating:

| True Light [coming] | Lumen verum |
| From the true Light, | de vero lumine, |
| The Word made flesh | Verbum caro |
| In the Virgin, | factum in virgine, |
| Then, the [Word] | Tunc, quod erat |
| who was God appeared | Deus in homine, |
| in man. | Aparuit.[26] |

*Hymnus 102. In Purificatione Beatae Mariae Virginis*, sings the human engendering of the Word/Light is sung through the words in these lyrical verses:

| You, who, when greeted, conceive the Word | Quae salutata aure concipis |
| Through the ear, engendering with flesh, | verbum, carne generans, |
| With a happy birth, from which | Felix partu, de quo es egressa |
| The most clear Light of the world came out. | mundi lux clarissima.[27] |

Gottschalk von Limburg (c. 1010–1098), in *Hymnus 284. (21.) De sanctibus. Iohanne Baptista et Iohanne Evangelista* formulates the virginal incarnation of God the Son, alluding to a well-known biblical passage in these verses:

| This Word, made flesh | Hoc verbum, caro factum |
| In a virgin mother, | in matre virgine, |
| Came out of her bed | processit de thalamo |
| As a kind husband | sponsus amabilis |
| In a garment of flesh. | in habitu carnis.[28] |

### 3.5. Twelfth-Century Hymns

From the twelfth century, we have found these eleven hymns referring to the subject:

*Hymnus 326. De conceptione sanctae Mariae virginis. Antiphona. In secundo nocturno* alludes to the virginal human conception of the Word/Light in the womb of Mary, reflecting it in the ancient prefiguration carried out by Gideon when he states:

| The Father's Word shone forth upon the world | Verbum patris mundo fulsit |
| Through the womb of the Virgin, | virginis per uterum, |
| Whose mind was not troubled | cujus mentem non gravavit |
| By the oppressive weight of sins, | onus premens scelerum, |
| As the rain on [Gideon's] fleece | sicut in vellus pluvia |
| So it descended upon Mary. | sic descendit in Maria.[29] |

*Hymnus 346. De beata Virgine Maria* expresses the virginal human engendering of the divine Word when Mary receives and accepts the heavenly word through these suggestive concepts:

| | |
|---|---|
| On behalf of the Creator | Ex parte rerum principis |
| The announcer [Gabriel] brings you salvation | salutem tibi nuntius |
| With the seed of the word, | affert semini verbius, |
| As you perceive the word by ear, | dum verbum aure percipis, |
| You conceive the Word by the word, | in verbo verbum concipis, |
| The Son of God becomes your son. | fit tuus dei filius.[30] |

*Hymnus 372. De Nativitate Domini, in gallicantu*, celebrates the human birth of the Word of God with these brief phrases:

| | |
|---|---|
| The heart of the Supreme Father | Eructavit cor |
| Exhaled the good Word, | summi patris verbum bonum, |
| Man ate | manducavit |
| The bread of angels | homo panem angelorum, |
| On this day. | die ista.[31] |

*Hymnus 375. Alia de sancta Maria (troparium)* alludes to the incarnation of the Word of God in this stanza:

| | |
|---|---|
| Preparing credulous hearts | Mox ad haec dicta |
| At these words | parans credula |
| You immediately conceive | corda concipis |
| The Lord Sabaot; | dominum sabaoth; |
| Thus the Word became flesh | sic verbum caro factum est |
| From you, holy Virgin. | ex te, virgo sacra.[32] |

*Hymnus 504. Psalterium Mariae* praises Mary for virginally giving birth to God the Son made man in these rhymes:

| | |
|---|---|
| Hail, whose understanding | Ave, cujus intellectum |
| Adjusted to the perfect | tunc instruxit ad perfectum |
| Word of God, when he took from you | verbum dei, quando carnem |
| Flesh not through flesh [intercourse]. | ex te sumpsit non per carnem.[33] |

*Hymnus 504. Psalterium Mariae* greets the Virgin for having made possible the human incarnation of the Word of God through these verses:

| | |
|---|---|
| Hail, whose Son is [the one] | Ave, cujus filius est, |
| Through whom God spoke, | per quem deus locutus est, |
| In which the Word, whom God begot, | in qua carne se induit |
| was clothed in flesh. | verbum, deus quod genuit.[34] |

*Hymnus 516. De sancta Maria* glorifies the Virgin for having conceived the Word of God after receiving his word through these consonances:

| Rejoice, gracious Virgin, | Gaude virgo gratiosa, |
| You conceived the Word with the word, | verbum verbo concepisti, |
| Rejoice, fruitful land, | gaude tellus fructuosa, |
| You brought forth the fruit of life. | fructum vitae protulisti.[35] |

*Hymnus 87.* In *Annuntiatione Beatae Mariae Virginis* also expresses the conception of the Word through the word in these brief verses:

| The Word is incarnated by the word | Verbum verbo incarnatur, |
| While Mary is greeted | Dum Maria salutatur |
| By the helper of salvation [Gabriel]. | A salutis bajulo.[36] |

*Hymnus 128. De beata Maria Virgine* enunciates the engendering of the Word through the divine breath in these terms:

| When [God] exhales his breath, | Ex Maria virgine |
| The Word of God begins to exist | Conspirante flamine |
| From the Virgin Mary. | Verbum Dei nascitur.[37] |

*Hymnus 154* urges us to celebrate the conception of the Redeemer in Mary's womb, stating:

| Let the plebs of the new fate, | Flore vernans gratiae |
| That sprout with the flower of grace, | Plaudat omnis hodie |
| Applaud today. | Turba novae sortis, |
| When the Word entered the Virgin, | Verbum intrans virginem |
| He restored man, | Restauravit hominem |
| When the power of death was destroyed. | Fracto jure mortis.[38] |

Dated circa the twelfth century, *Hymnus 331. De eadem [conceptione beatae Mariae Virginis], ad nonam hymnus* celebrates the birth of the Word/Light of God in this stanza:

| The Word made flesh is born into the world, | Nascitur mundo verbum caro factum, |
| A perfect light that surpasses the light of the sun, | solis transcendens lucem lux perfecta, |
| That shines in the darkness, which | tenebris lucens, capere quam sui |
| Those of his kind do not want to accept. | generis nolunt.[39] |

### 3.6. Thirteenth-Century Hymns

We have accredited these six hymns alluding to the subject under study, dating from the thirteenth century:

*Hymnus I. Psalterium beatae Mariae Virginis. Tertia Quinquagena* glorifies the virginal mother of the divine incarnate Word through these rhymes:

| Hail, whose heart | Ave, cuius exaltatum |
| Was not exalted nor ungrateful, | Cor non fuit nec ingratum, |
| When the Word is carried | Cum intra cubiculum |
| To you within the cubicle [womb], | Ad te verbum est delatum, |
| But the seal of virginity | Sed permansit illibatum |
| Remains unblemished. | Pudoris signaculum.[40] |

*Hymnus 77. De Beata Maria Virgine* praises Mary for having virginally conceived the incarnate divine Word through the word, by means of these polished rhymes:

| | |
|---|---|
| The Virgin is impregnated with the word, | Virgo verbo impraegnatur, |
| The Word of God becomes man | Verbum Dei humanatur |
| Not with the seed of a man. | Non virili germine. |
| A creature gives birth to the Creator, | Creatura creatorem |
| A Virgin [gives birth] to the Savior | Parit, virgo salvatorem |
| For our medication. | Nostro medicamine.[41] |

*Hymnus 118. De Beata Maria Virgine* praises Mary for having virginally conceived God the Son made man, saying:

| | |
|---|---|
| Give birth to the incarnate Word | Ut reformet mundi statum, |
| To reform the state of the world; | Verbum parit incarnatum; |
| Oh, what chaste fecundity, | O quam casta fecunditas, |
| And what fruitful chastity! | Et quam fecunda castitas! |
| A pure Virgin is fecundated, | Pura virgo fecundatur, |
| The true God becomes man, | Verus Deus humanatur, |
| Let the law be amazed before the Virgin mother, | Stupet lex matrem virginem, |
| Let the mind be amazed before a God-man. | Stupet mens Deum hominem.[42] |

*Hymnus 583. Sequentia* states that the Virgin conceived the Word of God upon receiving his word in these terms:

| | |
|---|---|
| He gave faith and obeyed, | Fidem dedit et obedit, |
| He believed the word, he gave birth to the Word, | verbo credit, verbum edit, |
| He yielded his mind and his belly to the Word | mentem ventrem verbo cedit, |
| As a pleasant lodging for the Pleasant One. | grato gratum hospitium.[43] |

*Hymnus 359. Ejusdem. [De sancta Maria]* glorifies the divine Word, incarnated as man in the virginal womb of Mary, when he states:

| | |
|---|---|
| The Supreme Height sent the law, | Misit legem specula superna, |
| The Eternal Vision [sent] the Word of peace, | verbum pacis visio aeterna, |
| A Word that an intact mother produced, | verbum, quod fudit mater intacta, |
| A Word by whom the world is made. | verbum, per quod saecula sunt facta. |
| This Word, made flesh | Hoc verbum in utero puellae |
| in the womb of a virgin bowed the heavens, | factum caro coelos inclinavit, |
| this one exalted the earth above the heavens | hoc ab impetu maris procellae |
| from the impetus of the tempest of the sea. | super coelos terram exaltavit.[44] |

*3.7. Fourteenth-Century Hymns*

We have documented the following eighteen dated in the fourteenth century hymns referred to the theme under analysis:

*Hymnus 82. De Beata Maria Virgine* thanks Mary for having given us the Redeemer through these stanzas:

| | |
|---|---|
| The Word of the Father incarnate | Verbum patris incarnatum |
| Given to us by you | Per te nobis est donatum |
| From virginal flesh. | Carne ex virgínea. |
| The Word, through whom heaven was made, | Verbum, per quod caelum factum, |
| Suffered for the guilt of the world | Passum ob mundi reatum |
| By his grace alone. | Sua sola gratia.[45] |

*Hymnus 384. De eadem [sancta Maria]* manifests, through the biblical prefiguration starring Gideon, the supernatural conception of God the Son in the virginal womb of Mary, with these consonances:

| | |
|---|---|
| Like dew on the grass | Sicut ros in gramine |
| The Word of the Supreme Father | descendit in virgine |
| Descended into the Virgin: | verbum summi patris: |
| [the Word] did not leave the Father, | Patrem non deseruit, |
| But assumed a mortal form | sed mortalem induit |
| in the mother's womb. | formam alvo matris.[46] |

*Hymnus 390. Dominica infra octavas Nativitatis Domini. Prosa* celebrates the supernatural birth of the divine Savior in these verses:

| | |
|---|---|
| Let the Church say a hymn | Ante thronum virginalem |
| For the spiritual world | hymnum dicat spiritalem |
| Before the virginal throne, | per orbem ecclesia, |
| On which the Word of the Father | in quo jacet, sicut placet, |
| Rests, as He pleases, preserved | verbum patris suae matris |
| The virginity of his mother. | salva pudicitia. |
| By this incarnate Word | Per hoc verbum incarnatum |
| The restored race of Adam | genus Adae reparatum |
| Returns to heaven. | redit ad coelestia.[47] |

*Hymnus 458. Gaudia beatae Mariae* congratulates the mother of the Word for having conceived him by receiving the word in this stanza:

| | |
|---|---|
| Rejoice, glorious Virgin, | Gaude virgo gloriosa, |
| You conceived the Word with the word, | verbum verbo concepisti, |
| Rejoice, fruitful earth, | gaude tellus fructuosa, |
| You brought forth the fruit of life. | fructum vitae protulisti.[48] |

*Hymnus 508. Roseum crinale beatae Virgine Mariae* greets the virgin mother of God the Son with these concepts:

| | |
|---|---|
| Hail, virginal star, | Salve virginale sidus, |
| Through you the faithful man lives, | per te vivit homo fidus, |
| Through you faith took root, | per te fides inolevit, |
| In you the Word of God grew, | in te verbum dei crevit, |
| And you did not know man. | virum nec scivisti.[49] |

*Hymnus 530. De eadem [beata Virgine Maria]. Sequentia* praises the Virgin for having engendered the divine Word by receiving the word through these brief verses:

| You conceived the Word through the word, | Verbum verbo concepisti, |
| You gave birth to the King of kings, | regem regum peperisti, |
| A virgin without knowing a man. | virgo viri nescia.[50] |

*Hymnus 13. De conceptione Beatae Mariae Virginis. In 2. Nocturno. Antiphonae* manifests the prodigious conception of the Word in the Annunciation with this stanza:

| Mary's heart breathed | Cor Mariae verbum bonum |
| Entirely the good Word, | Prorsus eructavit, |
| While the angel greeted her | Dum angelus per coeli deum |
| Through the God of heaven. | Eam salutavit.[51] |

*Hymnus 3. De sanctissima Trinitate* urges us to accept the virginal conception of the Savior through these suggestive expressions:

| It is necessary to believe | Patris verbum incarnatum |
| That the Word of the Father was incarnated | Sine patre matris natum, |
| And was born without a father from a mother. | Est necesse credere. |
| For in heaven [He exists] without a mother, | Nam in coelis sine matre, |
| But on earth, without a father, | Sed in terris sine patre |
| He comes to redeem us. | Nos venit redimere.[52] |

*Hymnus 71. De beata Maria Virgine* sings of the miraculous incarnation of God the Son in the Annunciation, expressing:

| The Word of the Father incarnate | Verbum patris incarnatum |
| Is given to us on our behalf | Per de nobis est donatum |
| By sublime glory, | Sublimi de gloria, |
| Assuming the form of his servant | Formam servi sumens sui |
| From a virginal flesh. | Carne ex virginea. |
| Gabriel indeed announced to you | Tibi quidem nuntiavit |
| And prophesied | Gabriel ac prophetavit |
| As these sacred mysteries, | Haec sacra mysteria, |
| Saying "Hail" so softly, | Dicens Ave tam suave, |
| Most fruitful with offspring. | Prole fecundissima.[53] |

*Hymnus 89. De beata Maria Virgine* exalts Mary's virginal divine motherhood with these far-fetched concepts:

| You conceived the Word of the Father, | Verbum patris concepisti, |
| As a daughter you gave birth to the Father, | Patrem nata peperisti |
| Bringing joy to the world. | Mundo ferens gaudia, |
| As daughter of the Son, and mother of the Father, | Nata nati, mater patris, |
| You have the name of mother wonderfully | Modo miro nomen matris |
| Without knowing man. | Habes viri nescia.[54] |

*Hymnus 103. Ad Beatam Mariam Virginem* asks for the intercession of the one who, welcoming the word, conceived the Word, in these brief verses:

| | |
|---|---|
| Hail, Mother of Jesus Christ, | Ave, mater Jesu Christi, |
| You conceived the Word with the word, | Verbum verbo concepisti, |
| Save me from a sad death. | Serva me a morte tristi.[55] |

*Hymnus 126* exalts the redemptive power of the incarnate God the Son through these poetic concepts:

| | |
|---|---|
| This Word incarnated in the womb of a virgin | Hoc verbum in utero puellae |
| Tilted the heavens, | Factum caro coelos inclinavit, |
| Thus exalted the earth above the heavens | Sic ab impetu maris procellae |
| from the force of the storm of the sea. | Supra coelos terram exaltavit.[56] |

*Hymnus 148. Super Ave Maria* proclaims the immaculate incarnation of the Word through the breath of God the Father, with these suggestive expressions:

| | |
|---|---|
| United with you in will, | Tecum iuncto numine |
| The Father with his breath | Verbum incarnavit |
| Incarnated the Word, | Genitor cum flamine, |
| Through whom he freed us | Per quod liberavit |
| From all sin, | Nos ab omni crimine |
| And consecrated you. | Teque consecravit, |
| The Sun prefigured | Quod lucens in virgine |
| What is shining in the Virgin. | Sol praefiguravit.[57] |

*Hymnus 88. De Beata Maria Virgine* exalts the supernatural human conception of the divine Word in the womb of Mary by stating:

| | |
|---|---|
| A child is born of a virgin, | Prodit puer de puella |
| The cell of virginity preserved, | Sana castitatis cella |
| Just as light comes from a star | Sicut exit lux de stella |
| And the ray [comes] from the Sun. | Et de sole radius. |
| The Word indeed entered [the womb] And | Verbum quidem introivit |
| the Word came out from there, | Atque verbum hinc exivit, |
| And so the womb that knew no man | Et sic laetus parturivit |
| Brought forth with joy. | Venter viri nescius.[58] |

*Hymnus 347. De sancta Maria* celebrates the supernatural conception of God the Son made man with these verses:

| | |
|---|---|
| The Word is united to the flesh | Verbum carni jungitur |
| In the womb of the Virgin, | virginis in utero, |
| And the [divine] nature of one | nec natura tollitur |
| Is not suppressed by the other [human]. | unius ab altero. |
| [. . .] | [. . .] |
| O wise child, | O puer sapiens, |
| O groaning Word, | O verbum vagiens, |
| O humble majesty! | O majestas humilis![59] |

*Hymnus 357. De beata Maria. Prosa* exalts the human generation of the divine Savior after the Virgin's unconditional acceptance of the Almighty's plan through these colorful elucidations:

| | |
|---|---|
| In her the great Word [stands], | Grande verbum in illa, |
| Who became the flesh of Christ, | quod factum est caro Christi, |
| As soon as the "let it be" [*fiat*] began, | Mox ut "fiat" incepit, |
| He conceived the true God | verum deum concepit |
| And man, | et hominem, |
| With whom he who deceived Eve | Quo, qui Evam decepit, |
| Found himself deceived by the Virgin. | deceptum se decerpit per virginem. |
| O word of nectar | O verbum nectareum, |
| By which the Virgin gave birth! | quo gignit virgo! |
| But, o virgin, who thus | sed o virgo, quae deum |
| Gives God to the world! | sic profert mundo![60] |

The Benedictine Engelbert of Admont (c. 1250–1331), abbot of the monastery of Admont in Styria, Austria, in his *Hymnus IX. Psalterium beatae Mariae Virginis. Prima Quinquagena* glorifies Mary for having conceived the Word of God by obediently receiving his word in this stanza:

| | |
|---|---|
| Hail, rose thus intact, | Ave, rosa sic intacta, |
| Transformed into a temple of truth, | Veritatis templum facta, |
| Throne of wisdom, | Thronus sapientiae, |
| You conceived the Word with the word, | Verbum verbo concepisti, |
| While you obeyed through faith | Dum per fidem paruisti |
| The inspiring grace. | Inspiranti gratiae.[61] |

The Cistercian Christian von Lilienfeld, Prior of Lilienfeld Abbey in Austria († 1330), requests in his *Hymnus 6. In Nativitate Domini* the redemptive intervention of the divine Word incarnated in Mary, when manifesting:

| | |
|---|---|
| Hail, Word exhaled | Ave, verbum eructatum |
| From the heart of the Most High, | De corde altissimi, |
| Word of God born of God, | Verbum Dei Deo natum, |
| From the most hidden source, | Fontis occultissimi, |
| You came from heaven, deliver | De caelo venisti, datum |
| A most splendid gift, | Muneris largissimi, |
| Cleanse me of sin | Mei absterge peccatum |
| From my most wicked heart. | Cordis iniquissimi. |
| [. . .] | [. . .] |
| Hail, Word incarnated | Ave, verbum incarnatum |
| from the Virgin Mary, | Ex Maria virgine, |
| Born not by the law of the flesh | Non de carnis iure natum |
| But by the holy Breath, | Sed sancto spiramine, |
| Behold me burdened | Intuere me gravatum |
| By the bond of sin, | Reatus ligamine, |
| From which return me freed | A quo redde liberatum |
| In your name. | Me in tuo nomine.[62] |

The Carthusian Conrad von Haimburg († 1360), Prior of the Carthusian monastery of Gaming in Austria, in his *Hymnus 463. Gaudia beata Virginis* celebrates Mary for having given birth to the Word made flesh after receiving the word of the angel at the Annunciation, with this stanza:

| | |
|---|---|
| Rejoice, Virgin, mother of Christ, | Gaude virgo, mater Christi, |
| You conceived the Word with the word, | verbum verbo concepisti, |
| While you heard from the angel: | dum ab angelo audisti: |
| Hail, full of grace! | ave plena gratia![63] |

Again, Conrad von Haimburg reiterates in *Hymnus 11. Oratio super Ave maris stella* similar concepts, celebrating Mary for having given birth to the divine Word after receiving the word of the heavenly herald, in these terms:

| | |
|---|---|
| Taking this "Hail," | Sumens illud Ave, |
| A word so sweet, | Verbum tam suave, |
| You [are] the dwelling place of Christ, | Tu Christi conclave |
| The portion of the chosen. | Electorum pars. |
| This Word given | Illud verbum datum |
| And born of the Father, | Et a patre natum, |
| Is engendered by you, | A te generatum |
| O sublime art. | O sublimis ars.[64] |

*3.8. Fourteenth- and Fifteenth-Century Hymns*

Dated at some imprecise date in the interval between the fourteenth and fifteenth centuries, we have documented these four hymns alluding to the topic under study:

*Hymnus 183. De X Gaudiis Beatae Mariae Virginis*, I celebrate Mary for having given birth to the divine Redeemer after accepting the word of the angel through these consonances:

| | |
|---|---|
| Rejoice, Virgin mother of Christ, | Gaude, virgo mater Christi, |
| While you received this "Hail," | Verbum verbo concepisti, |
| You conceived through the word the Word, | Dum hoc Ave suscepisti, |
| Who freed us from the sad Woe!; | Quod nos solvit a Vae tristi; |
| By this "Hail" so sweet | Per hoc Ave tam suave |
| Absolve us from the Woe! of death. | Nos absolve mortis a Vae.[65] |

*Hymnus XIV. Psalterium beatae Mariae Virginis. Prima Quinquagena* glorifies Mary for having conceived the Word of God by receiving his breath, with this stanza:

| | |
|---|---|
| Hail, whose lips | 47. Ave, cuius labia |
| Grace breathed, | gratia perflavit, |
| While the Word, | Dum, quod sapientia |
| Breathed | patris eructavit, |
| By the Wisdom of the Father, | Verbum tua viscera |
| thus consecrated your entrails | ita consecravit |
| and made there | Et pugnandi tegmina |
| the armor for fighting. | ibi fabricavit.[66] |

*Hymnus 1. De sancta Trinitate. In 2. Nocturno. Antiphonae* exalts the divine begetting of God the Son by the Father, reflecting it in the synonym Word = word, by stating:

| | |
|---|---|
| The Father breathes forth the Son | Eructat pater filium |
| As the mind [breathes forth] a good word, | Sicut mens verbum bonum, |
| Both being one principle | Ambo unum principium |
| Which breathes forth an eternal gift. | Aeternum spirant donum. |
| The Father, glory of all beings, | Pater cunctorum gloria, |
| By whom all things were created | Ex quo cuncta creantur |
| Through the Word, by whom | Per verbum, per quod omnia |
| All things are preserved. | Spiritu conservantur.[67] |

*Hymnus 4* manifests the human incarnation of the divine Word by the breath of the Father in these terms:

| | |
|---|---|
| Patent source of Paradise, | Paradisi patens fons, |
| You are Cypress, Mount Zion, | Tu Cupressus, Sion mons, |
| Bridge of sinners, | Peccatorum pons, |
| With the covering of the shadow of the Father | Patris obumbratione |
| The Word became flesh by you | Verbum caro fit per te |
| Through a sacred breath. | Sacro flamine.[68] |

*3.9. Fifteenth-Century Hymns*

Dated in the fifteenth century, we have cataloged these eighteen hymns referring to the subject under scrutiny:

*Hymnus 374. De sancta Maria et filio* expresses the engendering of the divine Word, who is born of Mary through the celestial word, in these verses:

| | |
|---|---|
| The chosen mother gives birth to the Word, | Mater electa dat verbum, |
| Who was made in her birth by the word, | quae nata facta per verbum, |
| The Word is born of God | de deo nascitur verbum |
| From the Virgin Mary. | Maria ex virgine.[69] |

*Hymnus 405. Visitationis Mariae, in primis vesperis,* exalts the conception of the divine Word in the womb of Mary, after accepting the Word of God transmitted by Gabriel, through this stanza:

| | |
|---|---|
| While she believes in her godfather [Gabriel], | Haec paranympho dum credit, |
| The sacred breath filled her, | sacrum hanc pneuma replevit, |
| Her womb swells and gestates | alvus tumescit et gerit |
| The Word of the Father, whom she deserved. | verbum patris, quod meruit.[70] |

*Hymnus 510. Ad beatam Mariam Virginem* celebrates the immaculate conception of the Word through the word by means of these poetic expressions:

| | |
|---|---|
| You were greeted thus: | Salutata sic fuisti: |
| "Hail, full of grace," | "ave plena gratia", |
| You conceived the Word through the word, | verbo verbum concepisti |
| A virgin without knowing man, | virgo viri nescia, |
| You gave birth to Jesus Christ, | Jhesum Christum peperisti, |
| Who washed all things, | qui lavavit omnia, |
| And after giving birth you remained | et post partum permansisti |
| A virgin always honest. | virgo semper integra.[71] |

*Hymnus 38. De corpore Christi. Ad Primam* sings the dual divine and human nature of the Word/Light of God in these verses:

| | |
|---|---|
| The Word proceeding from the Father, | Verbum prodiens a patre, |
| The Light of the true Light, | Lumen verum de lumine, |
| The God of heaven, humble man | Deus de coelis humilis |
| Visible to us. | Homo nobis visibilis.[72] |

*Hymnus 81. De Visitatione Beatae Mariae Virginis. Ad Laudes* expresses the conception of the Word/Light of God in the virginal womb of Mary by stating:

| | |
|---|---|
| The supreme Word proceeding | Verbum supernum prodiens |
| From the source of the living Light | A fonte vivi luminis, |
| Became a nascent man | Homo factus est oriens |
| Within the womb of the Virgin. | Intra uterum virginis.[73] |

*Hymnus 16. De praesentatione Beatae Mariae Virginis. In 3. Nocturno. Responsoria* recognizes that, upon receiving the word of the heavenly announcer, the ever-virgin Mary conceived the divine Word through these rhymes:

| | |
|---|---|
| When the angel announced it to you, | Verbum Dei concepisti |
| you conceived the Word of God, | Angelo annunciante, |
| whom you also gave birth to while being chaste, | Quod et casta genuisti |
| always a virgin, before and after. | Virgo semper, post et ante.[74] |

*Hymnus 5. De Sancto Spiritu. Ad Matutinum* asks for the grace of the Holy Spirit, who covered the Virgin in the Annunciation so that she might give birth to the incarnate divine Word, by stating:

| | |
|---|---|
| The Grace of the Holy Spirit | Nobis sancti spiritus |
| May it be given to us, | gratia sit data, |
| With which the Virgin of virgins | De qua virgo virginum |
| Was covered. | fuit obumbrata, |
| When she was greeted | Cum per sanctum angelum |
| By the holy angel, | fuit salutata, |
| The Word became flesh, | Verbum caro factum est, |
| The Virgin was fecundated. | virgo fecundata.[75] |

*Hymnus XIII. Psalterium beatae Mariae Virginis. Secunda Quinquagena* exalts Mary as the mother of the Word of God through this stanza:

| | |
|---|---|
| Hail, whose Son is | Ave, cuius filius est, |
| Through whom God has spoken, | Per quem Deus locutus est, |
| In whom the Word of God | In qua carne se induit |
| was clothed with flesh. | Verbum, quod Deus genuit.[76] |

*Hymnus 40. De Generatione Christi Aeterna et temporali* manifests the impeccable incarnation of the divine Word in these rhymes:

| | |
|---|---|
| Here is the Word of the Father, | Hic est verbum patrium, |
| The Word that configured | Verbum firmatorium |
| The heavens and the stars, | Caelorum et siderum, |
| The Word that became flesh, | Verbum, quod caro factum, |
| The Word that also fertilized | Verbum et quod intactum |
| An intact womb. | Fecundavit uterum.[77] |

*Hymnus 51. De Nomine Iesu* sings the supernatural human engendering of the Word of God, taking advantage of the synonymy Word=word, through these eloquent tropes:

| | |
|---|---|
| Jesus, the Word of the Father | Verbum patris genitum |
| Begotten from eternity, | Iesus ab aeterno |
| Weared the voice of the flesh | Vocem carnis induit |
| In the maternal womb, | in claustro materno, |
| The Word, most pure entity | Verbum ens mundissimum |
| In the supreme womb | in sinu superno |
| Becomes a voice as consonant | Fit tam vox quam consonans |
| In the interior of the womb. | in ventris interno. |
| The Word united to the model | Verbum iunctum plasmati |
| Is a simplified voice, | vox simplificata, |
| The Word united to man | Verbum iunctum homini |
| Is a duplicated voice, | vox est duplicata, |
| When conjugated with the Word | Tam carne quam anima |
| Both the flesh and the soul | verbo coniugata |
| The vowel becomes consonant | Vocalis fit consonans |
| Doubly combined. | duplo combinata.[78] |

*Hymnus 273. De Beata Maria Virgine* hails Mary for having given birth to the divine Word through the word of the Father transmitted by the archangel, recording:

| | |
|---|---|
| Hail, hail, Holy Mother, | Salve, salve, sancta parens, |
| Who, conceiving by the word of Gabriel, | Gabrielis verbo parens, |
| Conceives the Word with the word. | Verbum verbo concipis.[79] |

*Hymnus 104. Oratio de Virgine Maria, matre Jesu*, asks for the saving protection of Mary, whom it praises for having conceived the Word of God incarnate by accepting his word, through these consonances:

| | |
|---|---|
| Hail, Mother of Jesus Christ, | Ave, mater Jesu Christi, |
| You conceived the Word through the word, | Verbum verbo concepisti, |
| You gave birth to the true God, | Deum verum genuisti |
| Save us from the sad death. | Serva nos a morte tristi.[80] |

The famous Benedictine hymnographer Ulrich Stöcklins von Rottach, Abbot of Wessobrun Abbey in Bavaria between 1438 and 1443, in his *Hymnus 11. Oratio poenitentiae ad Christum et pro bono fine*, asks for the mercy of the divine Redeemer, made flesh in the womb of Mary, by expressing:

| | |
|---|---|
| Word, who received | Verbum in virgine |
| in the Virgin the flesh, | carnem suscipiens, |
| with which you forgave at death | Qua mundi scelera |
| the crimes of the world, | solvisti moriens, |
| I beseech you, be | Mihi, te deprecor, |
| merciful to me, | Esto compatiens, |
| because I am pursued | Quia persequitur |
| by the furious enemy. | me hostis saeviens.[81] |

Once again, Ulrich Söcklins von Rottach, in *Hymnus 25. Laudatorium Beatae Mariae Virginis. Ad Matutinum.* I salutes Mary for having made possible the human incarnation of the Word of God through these flowery tropes:

| | |
|---|---|
| Hail, o fig tree | Ave, o ficulnea |
| Which gives the sign of summer, | Signum dans aestatis, |
| When you surround the Word of the Father | Cum circundas trabea |
| With the purple mantle of flesh, | Carnis verbum patris, |
| O ivory house | O domus eburnea |
| Of the holy Trinity, | Sanctae trinitatis, |
| Lead me with the blessed | Ad verna aetherea |
| To the heavenly spring. | Duc me cum beatis.[82] |

Finally, Ulrich Söcklins von Rottach, in *Hymnus II. Psalterium Secundum. Quinquagena I*, turns to the saving protection of God the Father, begetter of his Word in this stanza:

| | |
|---|---|
| Oh, author of lights, | Auctor o luminum, |
| Almighty Father, | Pater omnipotens, |
| Who emits the ray | Emittens radium |
| And the wise Word, | Et verbum sapiens, |
| Teach | Cor meum stolidum |
| My foolish heart | Esto erudiens, |
| Which runs toward vice | Currens per vitium |
| Like a fool. | Velut insipiens.[83] |

### 3.10. Undated Hymns

From an imprecise date we have found these two hymns alluding to the subject under scrutiny:

*Hymnus 63. De Annuntiatione. Prosa* sings the virginal conception of the divine Word made flesh in the womb of Mary after the word of the angel, taking advantage of the rich homology between the terms Word [*Logos*] = word, in these complex rhymes:

| | |
|---|---|
| Gabriel was sent from heaven | Missus Gabriel de coelis, |
| As the faithful godfather of the Word | verbi bajulus fidelis, |
| To speak holy words | sacris disserit loquelis |
| With the blessed Virgin. | cum beata virgine. |
| The good and gentle Word | Verbum bonum et suave |
| Expandswithin the room, | pandit intus in conclave |
| And forming "Ave" of Eve | et ex Eva formans ave, |
| Inverting Eve's name. | Evae verso nomine. |

| | |
|---|---|
| [...] | [...] |
| Consequently, according to the covenant, | Consequenter juxta pactum |
| The Word made flesh presents itself, | adest verbum caro factum, |
| Yet the womb of the Virgin | semper tamen est intactum |
| Remains always intact. | puellare gremium.[84] |

*Hymnus 82. In Annuntiatione Beatae Mariae Virginis. Ad Laudes* sings the virginal human engendering of the divine Word in Mary through these brief verses:

| | |
|---|---|
| The Word that became flesh | Qui verbum, caro factus est, |
| With the announcement of the angel, | Praeconio angelico |
| Was born a virgin | De claustris virginalibus |
| From the virginal cloisters of the Virgin. | Virginis virgo natus est.[85] |

The exploration of an abundant corpus of patristic, theological, and hymnographic texts confirms plainly this fact: with greater or lesser poetic effectiveness, many of them exalted Mary for having virginally conceived the divine Word as a man after hearing and accepting "the word" (the message) of God the Father transmitted by Gabriel. It should not be forgotten, in fact, that Jn 1:1–4 had defined God the Son as the Word of God. In addition, since the Word of God wished to be incarnated in the virginal womb of Mary, she received this divine plan through the mouth of the archangel Gabriel, who communicated to her the word (the message) of God the Father. In fact, it was only at the very moment when the Virgin unconditionally accepted as a humble "handmaid of the Lord" (*ancilla Domini*) the word (the plan) of God that Gabriel transmitted to her that the Word of God was incarnated as man in her immaculate womb.

Therefore, numerous liturgical hymnographers studied here share the suggestive expression "You conceived the Word by the word" (*Verbum verbo concepisti*). This statement is equivalent to what the metaphor "conception by the ear" (*conceptio per aurem*) formulates, a special issue that is not possible to develop here and that we have already dealt with in another study (Salvador-González 2016b, pp. 83–122).

In conclusion, it is necessary to highlight the vast amount of patristic and theological exegeses and medieval liturgical hymn texts that so emphatically and concordantly proclaim the supernatural human conception/incarnation of the Word of God through the God's word (communicated by Gabriel) in the virginal womb of Mary. Therefore, it is surprising that this crucial Christological and Mariological theme has been ignored by renowned authors of books on Mariology (Paissac 1951; Alastruey 1952; Forte 1993; de La Potterie 1995; Bastero de Eleizalde 1995; Stock 1999; Fernández 1999; Ponce Cuéllar 2001; Cerbelaud 2005; Pozo 2005; De Fiores 2006–2008; Menke 2007; de La Soujeole 2007; Scheffczyk 2010; Calero de los Ríos 2010; Hauke 2021; García Paredes 2015; Casás Otero 2015; Bonarrigo 2018).

## 4. The Word's Incarnation in Some Paintings of the Annunciation from the Fourteenth, Fifteenth, and Sixteenth Centuries

We will now analyze eleven representations of the Annunciation to Mary in which we can see the expressive detail of God the Father emitting through His mouth—as if pronouncing a word or exhaling a breath—the beam of rays of light that falls on the head/ear of the Virgin Mary, carrying in its wake the Holy Spirit shaped as a dove. The presence of this bundle of rays of light coming from the mouth of God the Father, as a symbol of God the Son, could not be surprising if we remember that He, the Word of God, defined himself as "the Light of the world". So, in these eleven *Annunciations* that we will

now analyze, God the Father emits through His mouth, in the form of a ray of light, God the Son identified simultaneously as the Word and as Light.

Jacopo Torriti depicts *The Annunciation*, c. 1295—a scene that forms part of the mosaics Mariological iconographic program in the apse of the Basilica of Santa Maria Maggiore in Rome (Figure 1a,b—with great simplicity. On an abstract gold background, the artist places the minimal elements of the story, leaving out secondary details. With his feet on the ground and his wings spread out, the Archangel Gabriel blesses Mary, who stands upright before a luxurious canopied throne, with a suggestive appearance of a small temple or chapel. Opening her hands in front of her chest, the Virgin thus manifests her unconditional acceptance upon learning from the heavenly messenger God the Father's plan to make her the mother of God the Son incarnate without losing her virginity.

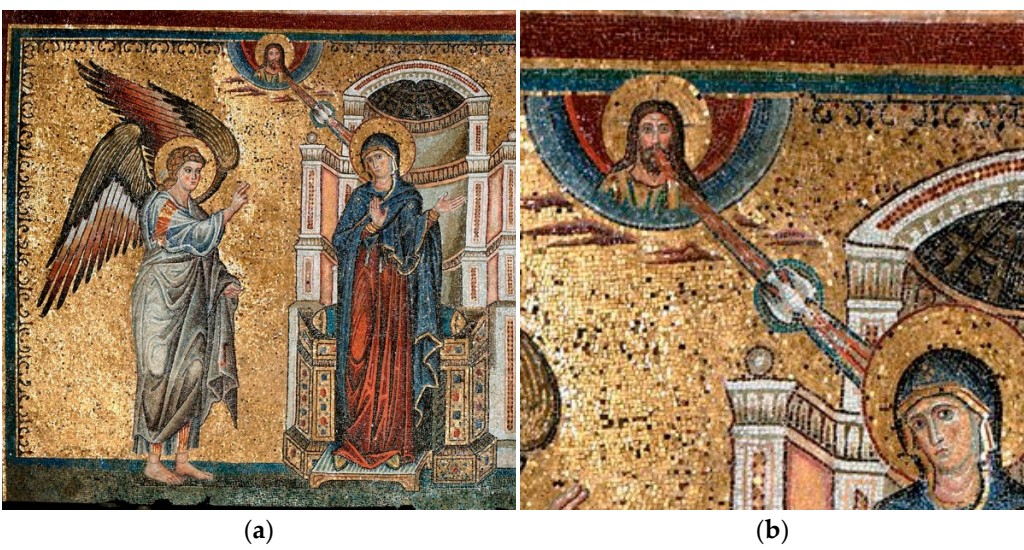

(**a**)          (**b**)

**Figure 1.** (**a**) Jacopo Torriti, *The Annunciation*, c. 1295. Basilica di Santa Maria Maggiore, Rome. (**b**) Detail from (**a**).

For the purposes of this research, the most interesting aspect of this mosaic is the fact that God the Father, shaped as a bust surrounded by a semicircular blue mandorla on the upper edge, emits from His mouth (as if exhaling a breath or pronouncing a word) towards the ear of the Virgin a beam of rays of light—a symbol of God the Son (Salvador-González 2020, pp. 334–55)—in whose wake flies the Holy Spirit figured as a dove.

An important detail must be pointed out about this character in the bust situated on the upper edge of this mosaic by Torriti: although His clear appearance as a young man would make Him identifiable with Christ (God the Son), in reality he represents God the Father, through whose mouth he emits the word/ray of light, which symbolizes the divine Word, Son of God. This physiognomic "slippage" is perfectly explainable, taking into account that the Christian are accustomed to designating the deity with the names "the Lord" and "God", without necessarily thinking of God the Father or God the Son; and, accustomed to the young physiognomy of Jesus Christ, it is not at all strange that in this mosaic the traditional figuration of God the Father as an old man has "slipped" here to that of a young man, as if he were the young Christ/God the Son.

With these suggestive details, the intellectual author of this *Annunciation*—probably Pope Nicholas IV himself, who commissioned Torriti this cycle of mosaics in the apse of the Basilica of Santa Maria Maggiore—wanted to visualize the supernatural human conception of the Word of God in her virginal womb at the very moment in which she heard/accepted His divine word transmitted by the archangel Gabriel.

In the end, in this mosaic of Torriti—as well in the other eleven *Annunciations* that we will analyze here—this beam of rays of light emitted as a word by the mouth of the eternal Father towards the Virgin's ear is a clear Christological/Mariological symbol: a symbol of the supernatural human incarnation of the divine Word in the immaculate womb of Mary, the virginal mother of this incarnate Son of God who will define himself in the Gospel of John with this the significant sentence "I am the light of the world".

The Master of the Madonna Strauss (active c. 1385–1415) structured *The Annunciation*, 1390–1395, from the Galleria dell'Accademia in Florence (Figure 2a,b) in a very simple way. In the left section of the painting, the Archangel Gabriel, holding in his left hand a symbolic lily stem, whose theological meanings we have explained in other articles (Salvador-González 2013, pp. 183–222; 2014, pp. 75–96; 2016a, pp. 117–44), kneels respectfully before the Virgin while blessing her with his right hand. Inside her home, summarized in a cubic room, Mary appears seated, holding the prayer book open on her lap, while she brings her right hand to her chest, as if in a gesture of surprise or questioning. Her womb appears swollen, a clear sign of having already conceived the incarnate Word of God at the very moment of having unrestrictedly accepted the plan of the Almighty, communicated as a word by the archangel. In this panel we can also see the ray of light, which, coming from God the Father (represented apparently as a young Christ in a tiny half-figure in the upper left corner), descends towards the Virgin, carrying in its wake the dove of the Holy Spirit.

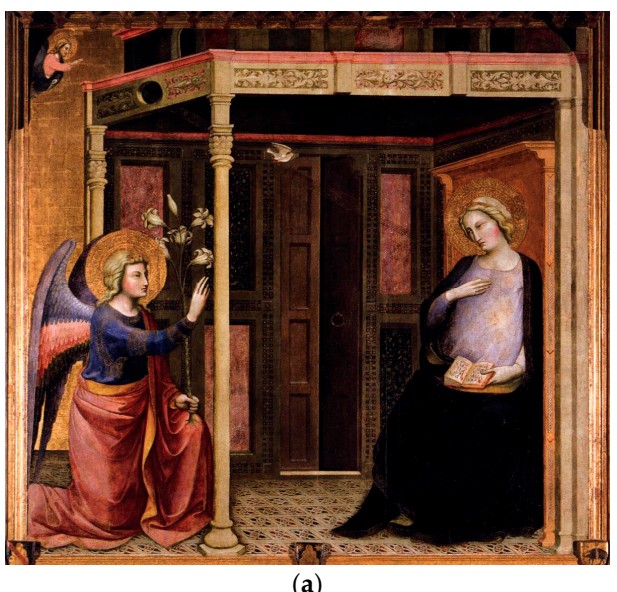
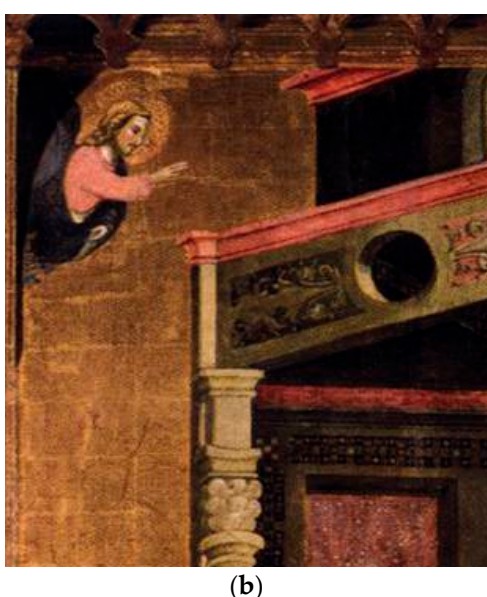

(**a**)　　　　　　　　　　　　　　　(**b**)

**Figure 2.** (**a**) Master of the Madonna Strauss, *The Annunciation*, 1390–1395. Galleria dell'Accademia, Florence. (**b**) Detail from (**a**).

However, the detail that is most motivating to highlight here is that this ray of light is emitted by the mouth of God the Father, as if it were a breath or, better yet, a word. As in the case of the Torriti's mosaic, such a significant circumstance in this panel can be interpreted as a clear allusion to the divine Word, who, engendered as God in eternity by the Father, now enters the virginal womb of Mary so that she may engender him as a man in time.

Melchior Broederlam (c. 1350–post 1409) designs a complex composition to depict *The Annunciation*, c. 1394–1399, in the left side of the left wing of *The Dijon Altarpiece (Retable of the Crucifixion)*, from the Musée de Beaux Arts in Dijon (Figure 3a,b). The painter shapes Mary's house as a vast, complex, temple-like architecture. Outside the garden protected by a crenellated wall—a symbol of the *hortus conclusus* (Salvador-González 2023, pp. 1–25; 2024, pp. 1–17)—Gabriel kneels before the Virgin beside a lily stem placed in a vase,

unfolding a phylactery on which is inscribed his initial greeting, *Ave gratia plena Dominus tecum*. Seated before the prayer book placed on a beautiful lectern, Mary raises her right hand in a gesture that could be interpreted as swearing the truth of a personal testimony or accepting someone else's proposal, in this case the plan of the Almighty announced by the heavenly messenger. Broederlam has especially highlighted here the wide jet of golden rays that, issued by the mouth of God the Father, falls on the Virgin's head/ear after passing through the glass of a window, without breaking or staining it. This feature is an effective metaphor to symbolize the supernatural human conception of God the Son in the virginal womb of Mary (Salvador-González 2022, pp. 390–85).

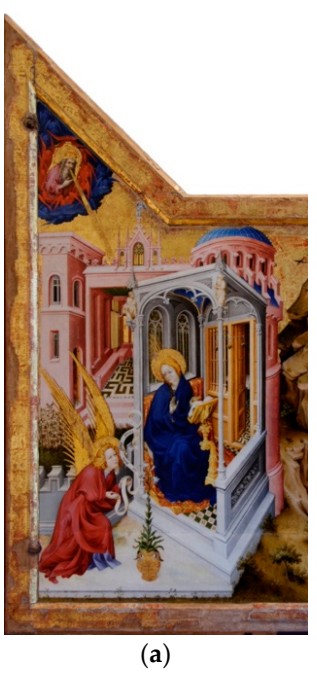 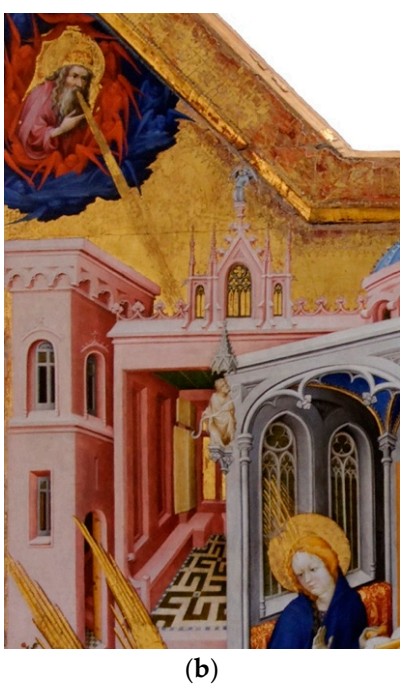

(**a**)  (**b**)

**Figure 3.** (**a**) Melchior Broederlam, *The Annunciation*, c. 1394–1399. Left side of the left wing of the *Dijon Altarpiece (Retable of the Crucifixion)*. Musée de Beaux Arts, Dijon. (**b**) Detail from (**a**).

However, the detail of greatest relevance for the purposes of this study is the fact that God the Father—a half-figure in a mandorla of cherubs and seraphim in the upper left of the painting—is emitting from his mouth this bundle of rays of light as if it were a breath or a word. With this precise detail, the intellectual author of this panel undoubtedly wants to symbolize that the eternal Father is sending his divine Word/Light to Mary so that she may engender him as a man in her virginal womb.

The Limbourg Brothers, in their *Annunciation*, c. 1412, on fol. 26r from *Les Très Riches Heures du Duc de Berry*, c. 1412–1416, at the Musée Condé in Chantilly (Figure 4a,b), depict the Virgin's house at Nazareth as an elegant temple in mixed style. Outside it, the archangel Gabriel, carrying the usual lily stem, kneels reverently before Mary while unfurling a banner bearing his eulogistic greeting. Inside this temple, kneeling on a prie-dieu before a prayer book, the Virgin turns her face slightly while raising her right hand in a double gesture similar to that depicted by Broederlam in the panel just analyzed. The Limbourg Brothers also include in this miniature, as Broederlam did in his panel, the beam of golden rays of light passing through the glass of a window before falling, together with the dove of the Holy Spirit, on the head/ear of the Virgin, thus symbolizing the same doctrinal meanings already explained in the previous painting. In a similar way to what Broederlam did, the Limbourg Brothers also depict here God the Father emitting through His mouth, as breath or word, the divine Word/Light in the form of golden rays towards Mary, to metaphorize the same dogmatic contents above.

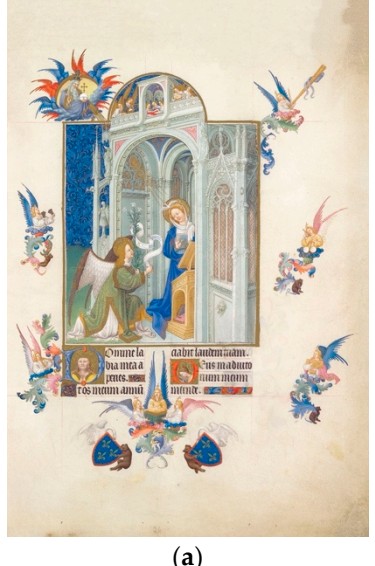
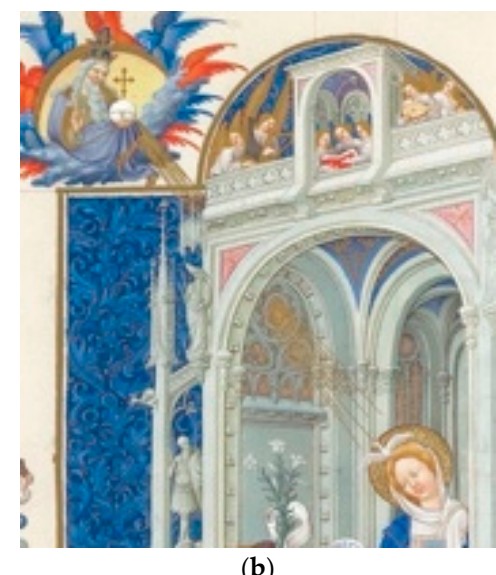

(**a**)          (**b**)

**Figure 4.** (**a**) The Limbourg Brothers, *The Annunciation*, c. 1412. Fol 26r from *Les Très Riches Heures du Duc de Berry*, c. 1412–1416. Musée Condé, Chantilly. (**b**) Detail from (**a**).

The very exceptional representation of *The Annunciation*, carved on the tympanum of the North portal of the Marienkapelle in Wurzburg, c. 1430 (Figure 5a,b), provides an ingenious solution to the problem under study. By distributing the constituent elements in a pyramidal form, perfectly adapted to the tympanum of the portal, the sculptor places at the base in a symmetrical arrangement, around the symbolic stem of lilies, the two genuflecting figures of Gabriel and Mary, the latter compensating for her mass/volume with an altar with two candelabras situated behind her. Crowning the compositional pyramid, God the Father appears enthroned in his mandorla of glory, holding in his left hand the sphere of the universe, while at his side two little flying angels lift a veil or curtain, perhaps with the purpose of metaphorizing the mystery of the human incarnation of God the Son that is being revealed (unveiled) in the episode of the Annunciation.

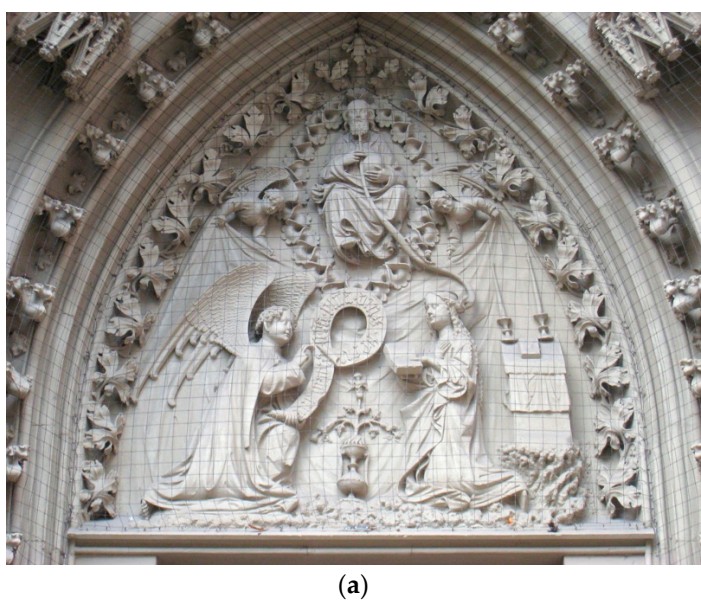
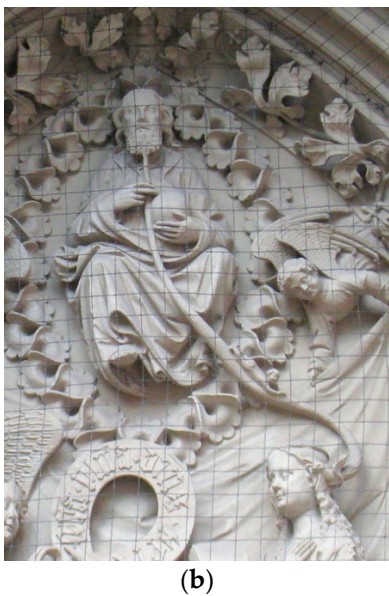

(**a**)          (**b**)

**Figure 5.** (**a**) *The Annunciation*, c. 1430, tympanum of the North portal of the Marienkapelle, Wurzburg (**b**) Detail from (**a**).

However, the most striking aspect of this high relief is the detail of God the Father blowing through a serpentine tube that ends in the Virgin's ear. Undoubtedly, the intellectual author of this sculptural *Annunciation* wanted to create a visual metaphor: the Almighty is breathing/pronouncing His divine Word engendered by Him in eternity towards Mary so that she may engender Him in time as a man through the word received by the ear, according to the thesis of the *conceptio per aurem*. Therefore, at the lower end of the tube, now touching the Virgin's ear, we can see the Holy Spirit depicted as a dove (*Spiritus Sanctus veniet in te*). That is why, even more surprisingly, a tiny, naked child/fetus (the Word of God) can be seen descending "outside" the tube, coming from the Father and entering Mary's virginal womb through her ear (*conceptio per aurem*).

The presence of this naked child/fetus (commonly called *homunculus*), almost always carrying a cross on his shoulder, in the scene of the Annunciation will be evident in some paintings and sculptures from Flanders, Netherlands, Italy, France, Spain, and Germany during the 14th to 16th centuries. Under the influence of the reformers, the Church prohibited its use in the 16th century; however, despite this prohibition, its inclusion can still be seen in some post-Reformation images. In any case, the study of this specific topic exceeds the limits of this article.

In this regard, we must point out that this "conception by the ear" should not be interpreted literally as a physical or real act, but rather as a metaphor, in the sense that Mary conceives the Word of God as a man when she hears and accepts the word of the Father proposing her to become the virginal mother of his divine Son incarnated as a man. In the same way, the fact that in this sculpture the fetus descends "outside" the tube does not mean that this tiny divine Word is alien or distinct from the word/breath emitted by the Father inside the tube: that "exteriority" is due to the fact that this is the only way the sculptor has of making the Word exhaled by God the Father toward the Virgin "visible".

Barthélemy d' Eyck (c. 1420–post 1470) staged *The Annunciation of Aix*, the central panel of *The Trypytich of the Annunciation*, 1443–1444, in the Église de la Madeleine in Aix-en-Provence (Figure 6a,b), in a monumental temple, a symbolic transmutation of Mary's humble home in Nazareth. Inside its naves, both protagonists appear kneeling face to face, dressed in ecclesiastical copes: Mary, opening her hands in a gesture similar to that of a priest officiating at Mass; the archangel, pointing with his left index finger to indicate to the Virgin that the Almighty has chosen her to become the mother of God the Son made man. In this painting too, we can see a beam of rays of light that, coming from the mouth of God the Father, passes through the panes of a circular window before falling on Mary's head/ear.

Barthélemy d'Eyck highlights the detail we are studying here. In the upper left corner of the panel, the divine Father, holding the sphere of the universe in his left hand and blessing Mary with his right, exhales towards her through his mouth, such as a breath or a word, the bundle of rays of light, symbolizing the Word of God/Light of the World. Not in vain does this Word/Light appear as a tiny fetus sliding through the rays of light shortly after they have passed through the window panes without breaking or staining them. All these are so many details that visually metaphorize the human conception incarnation of the Word of God in the virginal womb of Mary.

Benedetto Bonfigli (c. 1420–1496) staged *The Annunciation*, c. 1455, from the National Museum Thyssen-Bornemisza in Madrid (Figure 7a,b) in the courtyard or garden enclosed by the ornate walls of a luxurious palace, in which both protagonists remain kneeling facing each other. Carrying a stem of lilies in his left hand, Gabriel extends his right hand towards the Virgin, with his index and middle fingers extended in an attitude of blessing and indication, while greeting Mary with the AVE MARIA GRATIA PLENA, written in golden letters between both interlocutors.

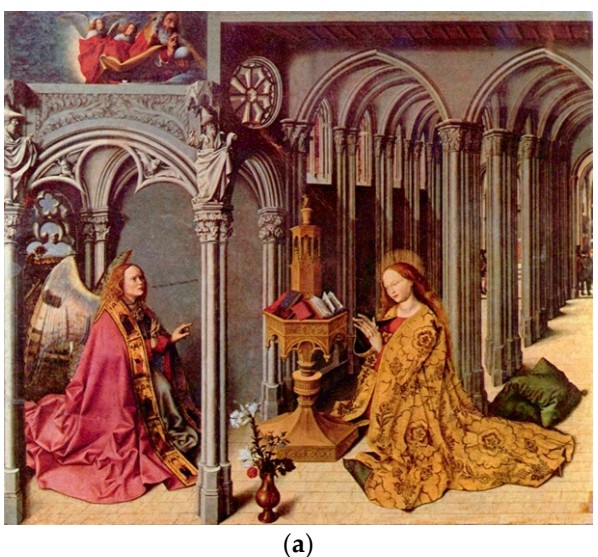 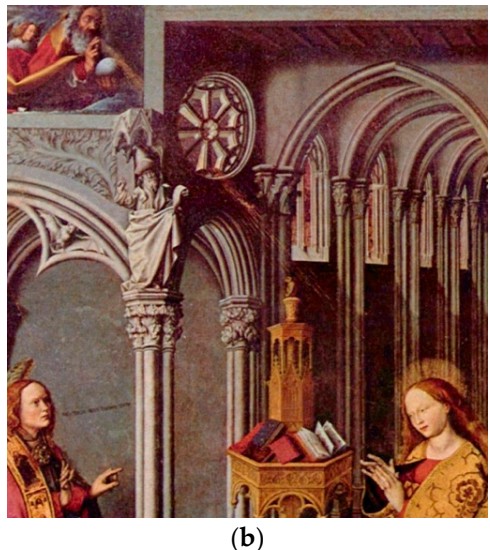

(**a**)         (**b**)

**Figure 6.** (**a**) Barthélemy d' Eyck, *The Annunciation of Aix*, central panel of *The Trypytich of the Annunciation*, 1443–1444. Église de la Madeleine, Aix-en-Provence. (**b**) Detail from (**a**).

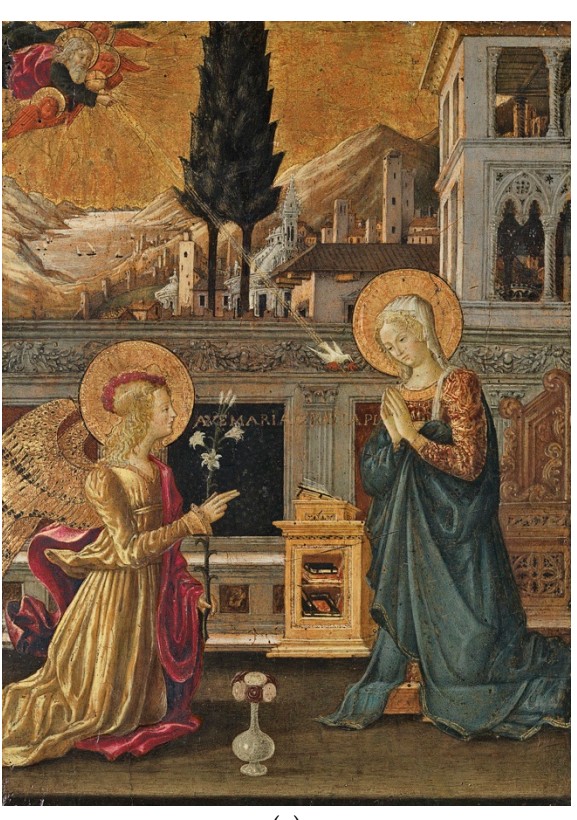 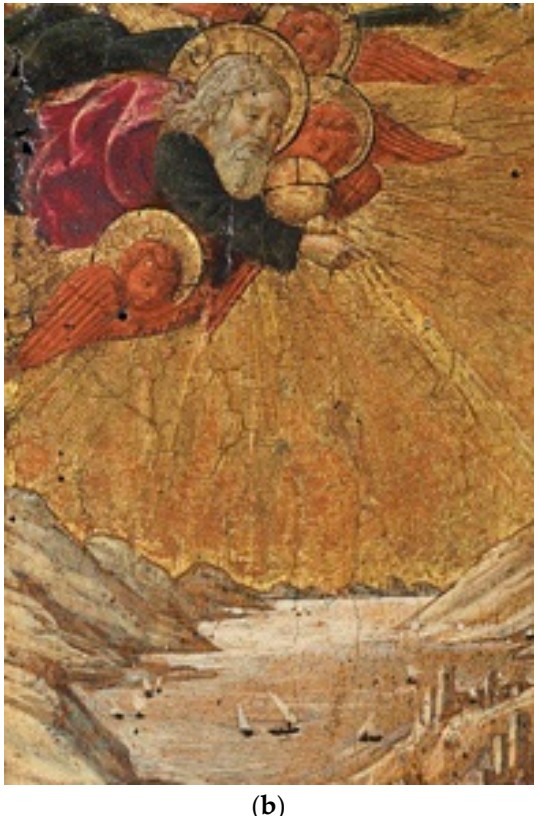

(**a**)         (**b**)

**Figure 7.** (**a**) Benedetto Bonfigli, *The Annunciation*, c. 1455. Museo Nacional Thyssen-Bonemisza, Madrid. (**b**) Detail from (**a**).

What is of interest here is the bundle of rays of light that God the Father, surrounded by cherubs and seraphim in the upper left corner, exhales through His mouth—as a breath or a word—toward the Virgin, with the dove of the Holy Spirit in His wake. In this way Bonfigli makes visible the consolidated doctrine of the Church according to which the Word of God, begotten by the Father in eternity, is virginally begotten by Mary as man in time, by the work of the Holy Spirit.

The Zaragozan brothers Nicolás and Martín Zahortiga (mid-fifteenth century) staged *The Annunciation*, c. 1460, from the Collegiate Church of Sainte Mary in Borja (Zaragoza) (Figure 8a,b) in a luxurious abode widely open to the landscape. The archangel, having just entered through the door, points with his right index finger toward God the Father while beginning to kneel before the Virgin. In his left hand he holds several lily stems, around which a phylactery is wound, where the salutation *ave gracia* [sic] *plena dom* is read in an inverted position (upside down). Kneeling before her prayer book, Mary opens her hands symmetrically before her chest—in a gesture similar to that of a priest officiating at mass—in an attitude of manifesting her unrestricted acceptance of the divine plan announced to her by the heavenly herald.

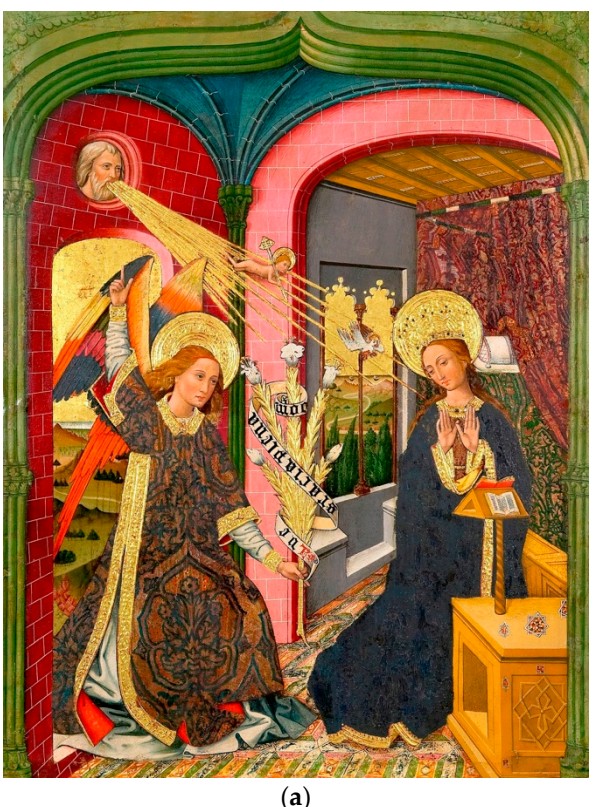
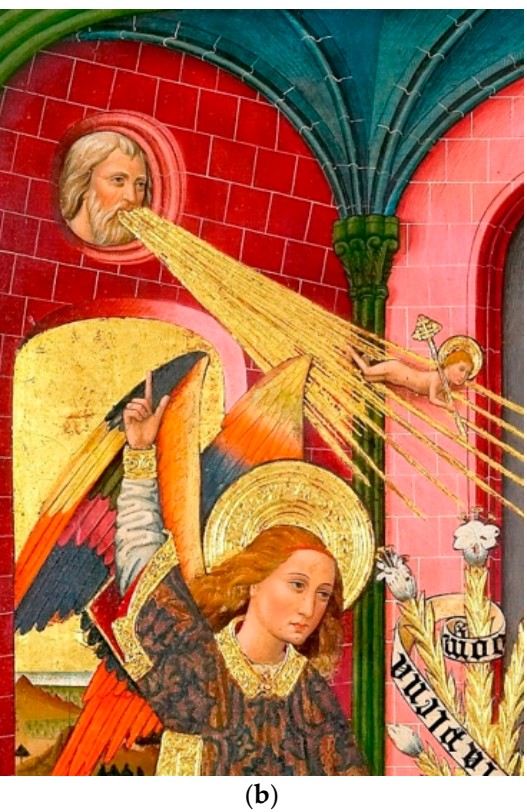

(**a**)    (**b**)

**Figure 8.** (**a**) Nicolás and Martín de Zahortiga, *The Annunciation*, c. 1460. The Collegiate Church of Sainte Mary in Borja. The Collegiate church of Borja Museum. (**b**) Detail from (**a**).

The most remarkable thing about this painting is that God the Father, whose figure, reduced to just his head, appears through a round window, emits from his mouth—as if exhaling a breath or pronouncing a word—a wide bundle of rays of light, in whose wake descends the "homunculus" God the Son with the cross on his shoulder, preceded by the dove of the Holy Spirit. This Word/ray of light that, with the little child Jesus, enters through Mary's ear—thus illustrating the thesis of the *conceptio per aurem*—visually metaphorizes once again the virginal conception/human incarnation of the Word of God in Mary's womb at the very moment in which she accepts unconditionally the divine word (message) transmitted to her by the archangel Gabriel.

Pedro Berruguete (c. 1450–1503) staged *The Annunciation*, c. 1490, from the Church of Sainte Eulalia in Paredes de Nava, Palencia (Figure 9a,b) in a clear space, barely furnished by a voluminous vase with some lily stems and the lectern on which the Virgin has her prayer book open. In this clear space the archangel, dressed in a wide cope, begins to kneel before Mary, while unfolding a long, serpentine banner on which is written his eulogistic greeting *Ave Maria gratia plena dns tecum bendic* [. . .]. Kneeling with her hands crossed over

her chest, the Virgin shows a gesture of shyness and modesty, while the divine beam of rays of light descends upon her with the dove of the Holy Spirit flying in it. It is motivating to highlight again the gesture of God the Father, who levitates in half figure in the upper left corner. Girded with the triple papal crown and holding in his left hand the sphere of the world, the Almighty exhales through his mouth his divine, eternal Word/Light shaped as a beam of rays toward the Virgin, so that she may engender him as a man through the Holy Spirit, who descends toward her (*Spiritus Sanctus superveniet in te*. Luke 1: 35) figured as a dove.

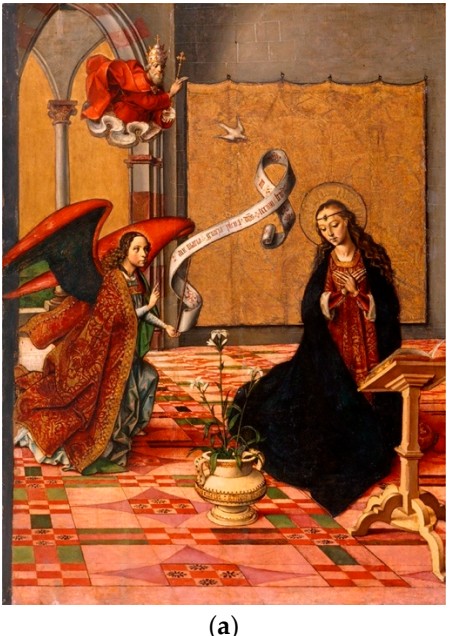 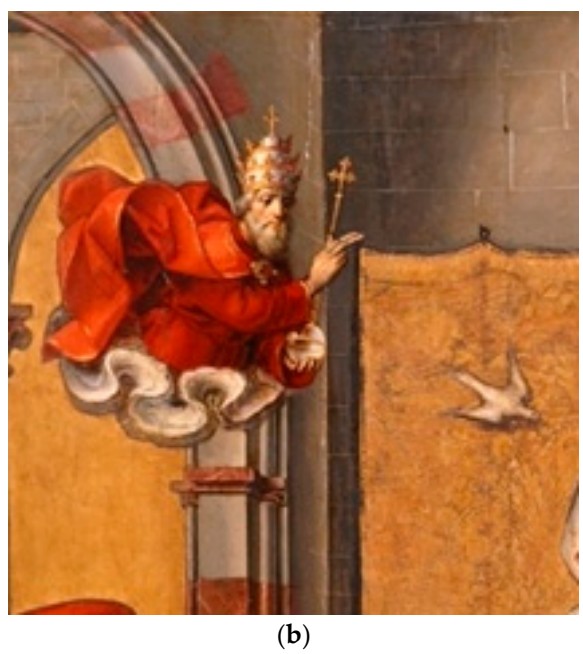

(**a**)                    (**b**)

**Figure 9.** (**a**) Pedro Berruguete, *The Annunciation*, c. 1490, Church of Sainte Eulalia, Paredes de Nava (Palencia). (**b**) Detail from (**a**).

Jacquelin de Montluçon (1467–1505) performed *The Annunciation*, c. 1496–1497, from the Musée des Beaux-Arts in Lyon (Figure 10a,b) in a kind of church nave with a shell-shaped quarter-sphere apse. Dressed in ecclesiastical vestments and holding a herald's golden scepter in his left hand, the archangel kneels before the Virgin while pointing with his right index finger to indicate his eulogistic greeting AVE MARIA GRATIA PLENA DNS TECUM, written in golden letters on a line that comes from his mouth towards Mary's ear. The painter here adds a touching detail: in the middle ground, two other angels lift the edges of the curtains of a canopy—in a situation analogous to the recently analyzed sculptural *Annunciation* at the Marienkapelle in Wurzburg—as to metaphorize the unveiling of the conception/incarnation of the Word of God in progress. However, the most significant factor in this painting for us is the fact that God the Father, levitating half-length outside the building in the upper left corner, exhales through his mouth, as if it were a word or a breath, a wide beam of rays of light toward the ear of the Virgin, carrying in his wake the dove of the Holy Spirit. It seems obvious to repeat that, with these features, the intellectual author of this painting wanted once again to metaphorically visualize the human engendering of the Word of God in Mary's virginal womb.

Alejo Fernández (c. 1475–1545) staged *The Annunciation*, c. 1508, from the Museum of Fine Arts in Seville (Figure 11a,b), in a luxurious Renaissance palace, widely open to the landscape. In this refined architectural space, the angel, with his herald's scepter, begins to kneel before the Virgin, while she, kneeling in prayer before her prayer book, modestly shows her unconditional obedience to the Almighty's will by lowering her eyes and crossing her hands over her chest. Apart from other conventional elements in this

Marian episode, it is stimulating to highlight in this work the circumstance that God the Father, represented half-length with his usual attributes in the upper left corner, emits from his mouth, as a word/breath, the bundle of rays of light towards the head/ear of the Virgin, so that the divine Word, engendered as God by the Father in eternity, would be engendered as man in time by Mary. To make this visual metaphor even more evident, Alejo Fernández captured in the wake of the bundle of rays of light a tiny fetus carrying a cross on its shoulders, a clear representation of the Word of God who becomes flesh in order to die on the cross to redeem mankind.

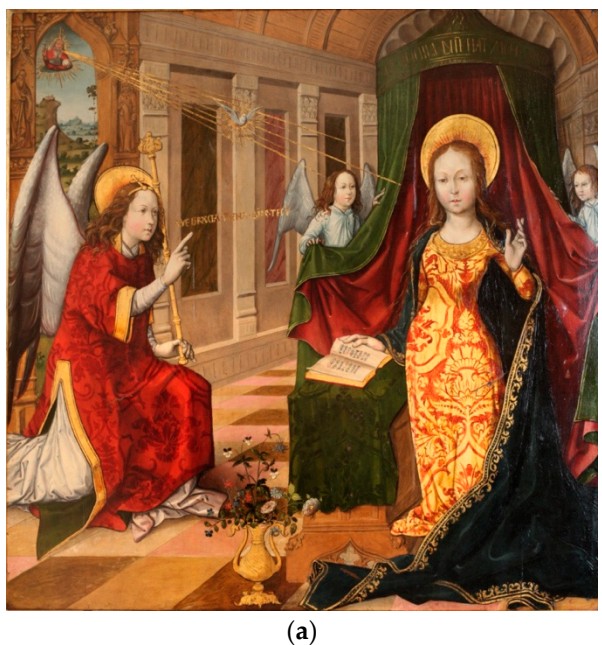
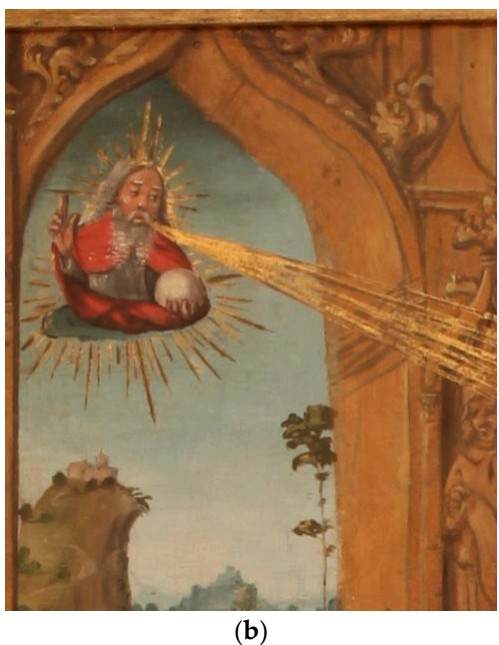

(**a**)        (**b**)

**Figure 10.** (**a**) Jacquelin de Montluçon, *The Annunciation*, c. 1496–1497. Musée des Beaux-Arts, Lyon. (**b**) Detail from (**a**).

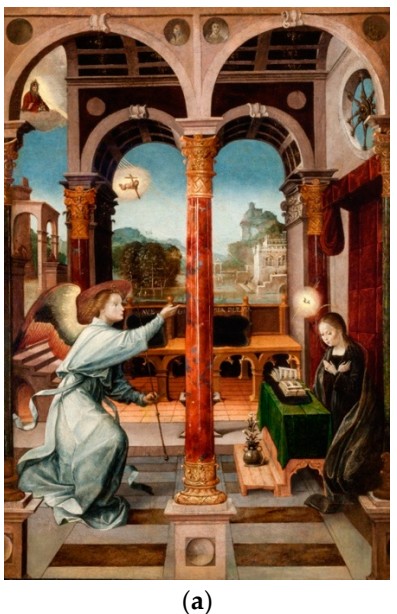
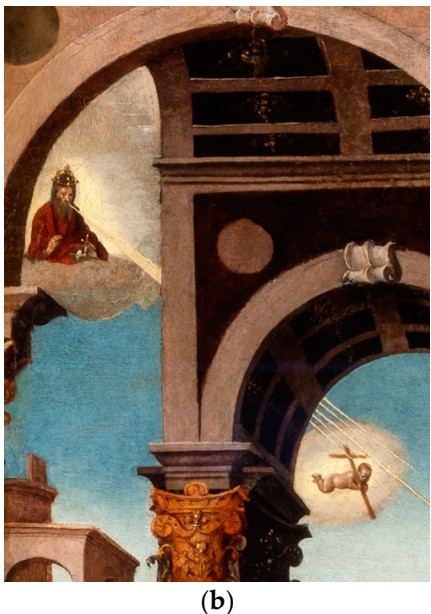

(**a**)        (**b**)

**Figure 11.** (**a**) Alejo Fernández, *The Annunciation*, c. 1508. Museum of Fine Arts, Seville. (**b**) Detail from (**a**).

The analysis of these eleven *Annunciations* reveals that, despite coming from very different socio-cultural ambits—Italy, Flanders, Netherlands, Spain, Germany, France—they

all coincide in symbolizing, through the visual metaphor of the beam of rays of light emitted by God the Father's mouth, the mystery of the human incarnation of God the Son, which numerous medieval Fathers, theologians, and liturgical hymnographers highlighted through the textual metaphor of the human engendering of the Word of God by the word in the virginal womb of Mary.

It is therefore surprising that this significant symbolism has been ignored by prestigious iconographers of Christian art (Mâle [1924] 1966, 1988; Bréhier 1928; Trens 1947; Panofsky [1953] 1966; Grabar 1979). On the contrary, other experts in the field, such as Louis Réau, Gertrud Schiller, and Giuseppe M. Toscano, mention this element in some way. Réau, after saying that in the images of the Annunciation the Holy Spirit descends in the form of a dove in the middle of a bundle of rays of light (Réau 1957, p. 185), specifies that in some *Annunciations* of the fifteenth century, you can see the surprising detail that from the mouth of God the Father comes a bundle of luminous rays, in whose wake descend the dove of the Holy Spirit and the Child Jesus, represented as a "homunculus" with a cross on his shoulder (Réau 1957, p. 190). For his part, Toscano begins by stating that in representations of the Annunciation, artists were quick to depict God the Father sending forth the dove of the Holy Spirit in a bundle of rays, before pointing out that "The rays sometimes come from the mouth of the Father, sometimes from his heart, sometimes from his hands". (Toscano 1960, vol. I, 1, p. 176). Toscano further indicates that, in Jacopo Torriti's *Annunciation* in the Basilica di Santa Maria Maggiore in Rome, God the Father sends forth the Holy Spirit from above, who flies forth in a bundle of rays coming from his mouth. (Toscano, vol. I, p. 178). Gertrud Schiller, commenting on Torriti's mosaic *Annunciation* in Santa Maria Maggiore, also points out that the bundle of rays of light in which the dove of the Holy Spirit flies emanates from God the Father, before rightly stating, "The fact that the ray of light in the mosaic proceeds from the mouth of God characterizes it as the word made flesh". (Schiller [1966] 1971, vol. I, p. 459).

One last clarification is necessary before concluding. In the analysis of several *Annunciations*, we mentioned the concept of the "intellectual author" of the painting or sculpture in the study. We formulate this concept logically, assuming that the painter or sculptor who created this artwork as the "material author" did not necessarily have to know the deep, subtle complexities of the Christological and Mariological dogmas underlying the bundle of rays of light emitted by the mouth of God the Father towards Mary. It is logical to think that the artist was instructed/required by the person who commissioned this special *Annunciation* (probably an ecclesiastical hierarch), or by some cleric or expert in theology delegated by the former, to represent the mystery of the virginal conception/incarnation of the Word of God through this suggestive visual metaphor of the ray of light exhaled by the mouth of the eternal Father. Furthermore, the concept of "intellectual author" remains valid even in the, quite frequent, case where an artist, in order to depict his *Annunciation*, was inspired (paraphrasing it) by the prototype or model depicted by a prestigious artist: in such a case, the eventual "intellectual author" of the original prototypical model would also be that of the resulting copy or paraphrase. Furthermore, it is evident that the few artists who, such as Fra Angelico, Fra Filippo Lippi, Fra Carlevare, Fra Bartolomeo, Lorenzo Monaco, and a few more, because of their condition as priests or friars, had a broad theological culture, are simultaneously the intellectual and material authors of their artworks.

## 5. Conclusions

The starting point for this article was a significant discovery during an assiduous research into medieval Marian iconography. Knowing that in the images of the Annunciation, an almost constant element is a bundle of rays of light (symbol of God the Son) descending

towards the Virgin Mary, we surprisingly discovered that in several *Annunciations* this bundle of rays is emitted by the mouth of God the Father, as if it were a word or a breath.

In order to try to find a satisfactory answer to the enigma of this bundle of rays of light exhaled by the mouth of the Almighty, we researched the primary biblical, patristic, theological, and liturgical sources of Christianity that inspire and underpin the images of Christian art.

In the biblical field, the key was given to us by three New Testament passages: the first is the testimony of the evangelist John, who calls Christ the divine Word that has been with God the Father from eternity and is the light of humanity; the second is Christ's own assertion when proclaiming himself the light of the world that enlightens the faithful; the third is the manifestation of unrestricted obedience on the part of the Virgin Mary, accepting as "the handmaid of the Lord" that it be done in her according to the divine word transmitted by the archangel Gabriel.

In the patristic and theological fields, the author deliberately restricted himself to a brief approach to some exegesis of Fathers and theologians of the Eastern and Western Churches who dealt with the essence of Christ as the Word of God made flesh in the womb of the Virgin Mary, knowing that practically all of them address the subject to a greater or lesser extent. The result of this brief foray into patristic and theological sources was to note the emphasis with which all the Fathers and theologians agree on this thesis: the human conception/incarnation of the Word of God occurred supernaturally, without manly intervention, in the virginal womb of Mary at the very instant in which she unconditionally accepted the divine plan announced to her by Gabriel.

In fact, many of the analyzed liturgical hymns praised Mary for having virginally conceived the divine Word incarnate after hearing and accepting "the word" (the message) of God the Father communicated by the archangel. Therefore, numerous liturgical hymnographers share the evocative expression "You conceived the Word by the word" (*Verbum verbo concepisti*), a statement equivalent to the metaphor "conception by the ear" (*conceptio per aurem*).

Regarding the images that illustrate the subject under study, the author analyzed eleven *Annunciations* from the thirteenth to sixteenth centuries in which God the Father emits from His mouth—as if pronouncing a word or exhaling a breath—the bundle of rays of light that falls on the head/ear of the Virgin Mary, carrying in its wake the Holy Spirit shaped as a dove.

In this sense, the comparative analysis between the doctrinal texts and this very special feature allowed us to conclude that the latter illustrate as visual metaphors the textual metaphors contained in the writings of the Fathers, theologians, and medieval liturgical hymnographers: this ray of light emitted by the mouth of God the Father, as if it were a word or a breath, can be interpreted as the metaphorization that the divine Word, begotten as God in eternity by the Father, now enters the womb of Mary so that she may virginally beget Him as a man in time.

**Funding:** This research received no external funding.

**Institutional Review Board Statement:** Not applicable.

**Informed Consent Statement:** Not applicable.

**Data Availability Statement:** No new data were created or analyzed in this study. Data sharing is not applicable to this article.

**Conflicts of Interest:** The author declares no conflict of interest.

## Abbreviations

The following abbreviations are used in this manuscript:

PL        Migne, Jacques-Paul (ed.), *Patrologiae Cursus Completus*, *Series Latina*, Paris, Garnier, 1844–1864, 221 vols.

PG        Migne, Jacques-Paul (ed.), *Patrologiae Cursus Completus*, *Series Graeca*, Paris, Garnier, 1857–1867, 166 vols.

AHMA    Dreves, Guido M., and Blume, Clemens. *Analecta Hymnica Medii Aevi*, Leipzig, R. Reisland, 1886–1922., 55 vols.

Mone    Mone, Franz Josef (ed.). *Hymni Latini Medii Aevi. Tomus Secundus. Hymni ad B.V. Mariam.* Friburgi Brisgoviae, Sumptibus Herder, 1854.

Q IX      *Doctoris Seraphici S. Bonaventurae Opera Omnia. Tomus IX. Sermones de Tempore, de Sanctis, de B. Virgine Maria,*

          *et de Diversis.* Ad claras Aquas (Quaracchi): Ex typographia Colegii S. Bonaventurae, 1882

## Notes

[1]    "Iterum ergo locutus est eis Jesus, dicens: Ego sum lux mundi: qui sequitur me, non ambulat in tenebris, sed habebit lumen vitae". (Io 8:12. *Biblia Sacra iuxta Vulgatam Clementinam*, 2005, p. 1050).

[2]    "In principio erat Verbum, et Verbum erat apud Deum, et Deus erat Verbum. Hoc erat in principio apud Deum. Omnia per ipsum facta sunt: et sine ipso factum est nihil, quod factum est. In ipso vita erat, et vita erat lux hominum: et lux in tenebris lucet, et tenebrae eam non comprehenderunt". (Io 1:1–5. *Biblia Sacra iuxta Vulgatam Clementinam*, 2005, p. 1042)

[3]    "Ecce ancilla Domini, fiat mihi secundum verbum tuum". (Lc 1,38. *Biblia Sacra iuxta Vulgatam Clementinam*, 2005, p. 1011).

[4]    Proclus Constantinopolitanus, *Oratio XV. In sanctam Pascha et in illud "In principio erat Verbum"*. pp. 65, 799.

[5]    Ibid.

[6]    Ibid., p. 803.

[7]    Andreas Cretensis, *Canon in B. Mariae Nativitatem*. pp. 97, 1323.

[8]    Andreas Cretensis, *Idiomela*. pp. 97, 1434.

[9]    Iohannes Damascenus, *Homilia in Sabbatum Sanctum.* pp. 96, 614.

[10]   Ibid.

[11]   Ambrosius Mediolanensis, *Veni Redemptor gentium*. Published also, with the title *Hymnus 8. (5.) In Nocte Natalis Domini*, in AHMA 50, p. 13.

[12]   "Ait de his beatus evangelista Joannes (Cap. 1): *In principio erat Verbum, et Verbum erat apud Deum, et Deus erat Verbum*. Et iterum: *Et Verbum caro factum est*". (Maximus Taurinensis, *Homilia X. De Nativitate Domini V.* pp. 57–243).

[13]   Ibid.

[14]   Anselmus Cantuariensis, *Psalterium Dominae nostrae (Pars I)*. pp. 158–1039.

[15]   "Fiat mihi de Verbo secundum verbum tuum. Verbum quod erat in principio apud Deum, fiat caro de carne mea secundum verbum tuum. Fiat, obsecro, mihi verbum, non prolatum quod transeat, sed conceptum ut maneat, carne videlicet indutum, non aere. Fiat mihi non tantum audibile auribus, sed et visibile oculis, palpabile manibus, gestabile humeris". (Bernardus Claraevalensis, *Super missus est Homiliae. Homilia IV*, 11. pp. 183–86).

[16]   "Nec fac mihi verbum scriptum et mutum, sed incarnatum et vivum: hoc est, non mutis figuris, mortuis in pellibus exaratum, sed in forma humana meis castis visceribus vivaciter impressum: et hoc non mortui calami depictione, sed sancti Spiritus operatione". (Ibid.).

[17]   "Venit etiam Verbum a se, descenditque sub se, quando *caro factum est, et habitavit in nobis* (Joan. 1), quando semetipsum exinanivit formam servi accipiens (Philip. II). Illa exinanitio ejus descensio fuit. Ita tamen descendit, ut sibi non deesset. Ita *caro factum est*, ut Verbum esse non desineret, nec minuit gloriam majestatis, humanitatis assumptio". (Amadeus Lausannensis, *Homilia III. De Incarnatione Christi, ex Virginis conceptione de Spiritu Sancto.* pp. 188–1315).

[18]   "*Verbum caro factum est*, Ioannis primo'. Exprimitur in his verbis istud caeleste mysterium et admirabile sacramentum, istud opus magnificum et beneficium infinitum, quod Deus aeternus, humiliter se inclinans, limum nostrae naturae in suae assumsit unitatem personae. Tangitur ergo assumens nomine Verbi, assumptum nomine carnis, et ipsa assumtio et copula nomine factionis". (Bonaventura de Balneoregio, *In Nativitate Domini. Sermo II*: Q IX, 106b).

[19]   "Cum enim verbum mentale exterius profertur, quasi voce vestitur, et vox quidem ista progreditur, sonat in publicum, ut signatum maneat in occulto, quia vox percipitur sensu, signatum vero percipitur intellectu. Sed Verbum Patris prius quidem fuit nudum, quia nulli creaturae unitum; postea vero carne vestitum, ostendit exterius carnem, celans intra Divinitatem; Isaiae quadragesimo quinto: *Vere tu es Deus absconditus*. — Nota etiam, quod verbum mentis et verbum vocis non sunt duo verba, sed

unum, prius quidem nudum, postea vestitum. Sic Verbum-caro, cum sit Deus et homo, non duo sunt, sed unus est Christus".
(Ibid., 107a-b).

[20] Magnus Felix Ennodius, *Hymnus 63. (10.) Hymnus sanctae Mariae*. AHMA 50, p. 68.

[21] Rabanus Maurus, *Hymnus 145. (14.) Item de Nativitate Domini*. AHMA 50, p. 195.

[22] *Hymnus 2. De Adventu DN*. AHMA 14, p. 18.

[23] *Hymnus 11. De Annuntiatione B.V.M*. AHMA 2, p. 154.

[24] *Hymnus 6. Purificatio*. AHMA 2, p. 126.

[25] *Hymnus 87. De Nominibus Domini*. AHMA 53, p. 152.

[26] *Hymnus 9. In Epiphania Domini*. AHMA 40, p. 28.

[27] *Hymnus 102. In Purificatione Beatae M. V*. AHMA 53, p. 177.

[28] Godescalcus Lintpurgensis, *Hymnus 284. (21.) De ss. Iohanne Bapt. et Iohanne Evang*. AHMA 50, p. 366.

[29] *Hymnus 326. De conceptione s. Mariae virg. antiphona. In secundo nocturno*. Mone, p. 10.

[30] *Hymnus 346. De b. v. Maria.* Mone, p. 35.

[31] *Hymnus 372. De nativitate domini. in gallicantu*. Mone, p. 65.

[32] *Hymnus 375. Alia de s. Maria (troparium)*. Mone, p. 69.

[33] *Hymnus 504. Psalterium Mariae*. Mone, p. 235.

[34] *Hymnus 504. Psalterium Mariae*. Mone, p. 236.

[35] *Hymnus 516. De s. Maria*. Mone, p. 300.

[36] *Hymnus 87. ln Annuntiatione BMV*. AHMA 10, p. 75.

[37] *Hymnus 128. De beata Maria V*. AHMA 10, p. 100.

[38] *Hymnus 154*. AHMA 20, p. 122.

[39] *Hymnus 331. De eadem [conceptione b. Mariae v.], ad nonam hymnus*. Mone, p. 18.

[40] *Hymnus I. Psalterium beatae Mariae V. Tertia Quinquagena*. AHMA 36, p. 24.

[41] *Hymnus 77. De Beata Maria V*. AHMA 37, p. 77.

[42] *Hymnus 118. De Beata Maria V*. AHMA 40, p. 114.

[43] *Hymnus 583. Sequentia*. Mone, p. 397.

[44] *Hymnus 359. Ejusdem. [De s. Maria]*. Mone, 52. Published also, with a little variant, with the title *Hymnus 126*, in AHMA 20, p. 107.

[45] *Hymnus 82. De Beata Maria V*. AHMA 42, p. 90.

[46] *Hymnus 384. De eadem [s. Maria]*. Mone, p. 78.

[47] *Hymnus 390. Dominica infra octavas nativ. domini. prosa*. Mone, p. 87.

[48] *Hymnus 458. Gaudia b. Mariae.* Mone, p. 169. In the same terms, except for a change in an adjective ("graceful" instead of "glorious"), we can read *Hymnus 516. De s. Maria*, in Mone, p. 300.

[49] *Hymnus 508. Roseum crinale b. v. Mariae*. Mone, p. 275.

[50] *Hymnus 530. De eadem [b. virg. Maria]. sequentia*. Mone, p. 317. Published also, with the title *Hymnus 550. De Beata Maria Virgine*, in Mone, 351; and, with the title *Hymnus 280. De Beata Maria V.*, in AHMA 54, p. 424.

[51] *Hymnus 13. De conceptione BMV. In 2. Nocturno. Antiphonae*. AHMA 5, p. 52.

[52] *Hymnus 3. De sanctissima Trinitate*. AHMA 8, p. 12.

[53] *Hymnus 71. De beata Maria V*. AHMA 8, p. 64.

[54] *Hymnus 89. De beata Maria V*. AHMA 8, p. 73

[55] *Hymnus 103. Ad B. Mariam V*. AHMA 15, p. 129.

[56] *Hymnus 126*. AHMA 20, p. 107.

[57] *Hymnus 148. Super Ave Maria*. AHMA 30, p. 269.

[58] Christianus Campoliliensis, *Hymnus 88. De Beata Maria V.* AHMA 37, p. 83.

[59] *Hymnus 347. De s. Maria*. Mone, p. 37.

[60] *Hymnus 357. De b. Maria. prosa*. Mone, p. 49.

[61] Engelbertus Admontensis, *Hymnus IX. Psalterium beatae Mariae V. auctore Engelberto Admontensi. Prima Quinquagena*. AHMA 35, 125.

[62] Christianus Campoliliensis, *Hymnus 6. In Nativitate Domini*. AHMA 41, p. 106.

[63] Conradus Gemnicensis, *Hymnus 463. Gaudia b. virginis*. Mone, p. 174.

[64] Conradus Gemnicensis, *Hymnus 11. Oratio super Ave maris stella.* AHMA 3, 40. Published also, with the title *Hymnus 498. Oratio super Ave maris stella*, in Mone, p. 220.

[65] *Hymnus 183. De X Gaudiis BMV*. I. AHMA 31, p. 186.

[66] *Hymnus XIV. Psalterium beatae Mariae V. Prima Quinquagena*. AHMA 35, p. 204.

[67] *Hymnus 1. De s. Trinitate. In 2. Nocturno. Antiphonae*. AHMA 24, p. 14.

[68] *Hymnus 4*. AHMA 1, p. 49.

69    *Hymnus 374. De s. Maria et filio*. Mone, p. 68.

70    *Hymnus 405. Visitationis Mariae, in primis vesperis*. Mone, p. 116.

71    *Hymnus 510. Ad b. Mariam v*. Mone, p. 284.

72    *Hymnus 38. De corpore Christi. Ad Primam*. AHMA 4, p. 30.

73    *Hymnus 81. De Visitatione BMV. Ad Laudes*. AHMA 4, p. 53.

74    *Hymnus 16. De praesentatione BMV. In 3. Nocturno. Responsoria*. AHMA 5, p. 61.

75    *Hymnus 5. De sancto Spiritu. Ad Matutinum*. AHMA 30, p. 15.

76    *Hymnus XIII. Psalterium beatae Mariae V. Secunda Quinquagena*. AHMA 35, p. 192.

77    *Hymnus 40. De Generatione Christi Aeterna et temporali*. AHMA 46, p. 61.

78    *Hymnus 51. De Nomine Iesu*, I. AHMA 46, p. 78.

79    *Hymnus 273. De Beata Maria V*. AHMA 54, p. 415.

80    *Hymnus 104. Oratio de V. M. matre Jesu*. AHMA 15, p. 130.

81    Udalricus Wessofontanus, *Hymnus 11. Oratio poenitentiae ad Christum et pro bono fine*. AHMA 6, p. 42.

82    Udalricus Wessofontanus, *Hymnus 25. Laudatorium B.V.M. Ad Matutinum*. I.AHMA 6, p. 88.

83    Udalricus Wessofontanus, *Hymnus II. Psalterium Secundum. Quinquagena I. AHMA 38, p. 28.*

84    *Hymnus 63. De annuntiatione. prosa*. Mone, pp. 55–56.

85    *Hymnus 82. In Annuntiatione B. M. V. Ad Laudes*. AHMA 27, p. 118.

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
