# Peer review of "Verbum Verbo Concepisti. The Word’s Incarnation in Some Images of the Annunciation in the Light of Medieval Liturgical Hymns"

_religions, doi:10.3390/rel16040456_

Round 1
Reviewer 1 Report
Comments and Suggestions for Authors
This is an excellent work that makes a great effort in terms of documentation, drawing on ancient and medieval sources to support the commentary on a series of works of art that the author analyses in section 4 with great detail and professionalism. This double aspect of the work is of great interest because it enriches the study of art with textual foundations that are often unknown or little known to art historians.
Despite the high quality of the article, I provide first some comments on its content. I will make some more technical or presentational comments later.
The article seems to focus too much on listing texts of Christian authors and hymns. Although this has great documentary value, there is a lack of a reasoned compilation of the data and an argumentation that gathers important information from these data for the exposition that follows in section 4. In fact, this documentation should be used in a more concrete way to justify the symbolic usage of some items that we find in the iconography studied in section 4. For example, the following question should be answered: which patristic or medieval texts justify the combination of the motif of ‘light’ with the motif of the ‘word’ that we find in the iconography (rays of light[?] coming from the mouth of the Father)? Is it really necessary to go to medieval texts or is the text of Jn 1 itself sufficient to explain it, since Jn 1 already says that ‘the Word is light’? If the text of Jn 1 is sufficient, why is it convenient to read the ancient and medieval texts, and what is new about them compared to the text of Jn 1 alone?
To make the need for argumentation in this regard clearer, I provide another example. The author emphasises the idea of the ‘conception by hearing’ which has a plastic representation in art: certainly the texts make mention of the Virgin's hearing. This is an example of an iconographic motif that would justify using the texts. But I don't think this detail is emphasised either when it appears in the texts or when the author is commenting the images. The comparison between the images and the texts in this regard seems only appear in the conclusions. However, this comparison should have been made explicit beforehand.
Another topic that could be object of consideration: the texts do not seem to make explicit other visual details that we find in the iconography. For example, in the selected texts there is no reference to the mouth of the Father (Christ is mentioned as the ‘mouth’ [of the Father?] and Gabriel’s mouth is mentioned: lines 186–88, but any mention of the mouth of the Father can be found): nevertheless, in the iconography we have repeatedly found the presence of the mouth of the Father as the origin of the rays of the light of the Word. It would therefore have been interesting to quote, for example, the 10th century hymn ‘Verbum salutis omnium Patris ab ore prodiens, Virgo beata, suscipe casto, Maria, viscere’ (AHMA 14, 18). Here we have a true mention of the mouth of the Father.
A last comment on the content. The mention of the ‘intellectual author’ of the works of art in the conclusions should be included in section 4 of the article. The conclusions of an article do not seem a good place for such clarifications.
Now, some comments on the presentation and other technical issues.
In the current template by Religions, there is a list of abbreviations to be used for an article. In this article there are abbreviations such as PL/PG (Patrologia Latina/Greca) or AHMA (Analecta Hymnica Medii Aevi) as well as some abbreviations for Bonaventura. An abbreviation list is highly recommended for an article like this one.
Footnote 1: it would be good to eliminate the number of the verse in the quotation (“12iterum…). The same could be said of:
- lines 39–44
- footnote 2
Notice the size of the text quoted among the following lines [by the way, in these quotations, there are heading numbers in many of the paragraphs that probably should be eliminated]:
- 113–14
- 121–22
- 175–76
- 179–80
- 184–85
- 186–348
Footnote 11: the author uses Spanish (instead of English) to do an explanation
Footnote 14: a mistake in the word Psalterium
Line 120: St Anselmus could be quoted by the critical edition by Schmitt
Line 122: St Bernardus could be quoted by the critical edition bt Leclercq
Another minor weakness of the article is that it has little secondary literature. There is a lack of bibliography on medieval theology on the topics addressed in the article, and the works cited are not very recent (for example, no works from the 21st century are cited).
Author Response
Please see the PDF below

Reviewer 2 Report
Comments and Suggestions for Authors
The fact that God the Father is represented as looking like Christ in figures 1, 2 should be addressed – is this a problem for the thesis or does it strengthen it? In most of the images discussed he is clearly represented, as per tradition, as an old man, so this could represent a significant iconographical difference.
There is no mention of the widely-known example of a homunculus/baby-carrying-a-cross appears in Campin’s Merode Altarpiece, which seems like a significant oversight. Campin does not include an image of God the Father, the relevance of which should also be addressed. Most homunculus imagery seems to have been created in the Netherlands, but although images from various European regions are presented, there is no discussion of regional differences or of why some regions would be more prone to including this imagery in Annunciations than others.
There is a body of secondary source material that is completely neglected, most of which concerns the rejection of the use of the homunculus in Annunciation imagery by 16th-century reformers (e.g., Johannes Molanus). Although Molanus’s views post-date the material presented in this article, these secondary sources do address the phenomenon to various extents, and the exclusion of this scholarship seems either purposeful or indicative of an authorial blind spot. (Actually, addressing counter-reformation objections could make for a much more nuanced study.)
References to the authors and their work -- e.g., "our careful study," "we were intrigued" -- should be excised from the article.
Comments on the Quality of English LanguageThe English is often too colloquial and a bit awkward. The first sentence of the abstract does not make sense grammatically or otherwise.
Author Response
See the PDF below
